# Thermoelectric device for active thermal concealment, deception and messaging

Yue Hou [1,2,6], Xiaosa Liang[1,6], Zhaoyu Li[3,6], Qianfeng Ding[1], Zheng Zhu[1], Xiaolong Sun[1], Chang Li[1], Wenjie Zhou[1], Wei Cao[1], Yuan Yu [4] ✉ & Ziyu Wang [1,2,5] ✉

Thermal concealment is vital for minimizing the visibility of individuals and vehicles to contemporary infrared surveillance technologies. Traditional approaches, such as emissivity modulation, are effective only in scenarios where the ambient temperature is lower than that of the target and typically exhibit response times on the order of minutes. Other temperature regulation methods generally operate within a restricted temperature range. This work presents an active thermal concealment cloak based on thermoelectric devices, integrating functionalities of infrared camouflage, deception, and information display. By optimizing the circuit design and incorporating a low-reflectivity black porous Ethylene-Vinyl Acetate film, the cloak achieves a uniform temperature distribution, eliminating distinct cold or hot boundaries, and exhibits strong resistance to light interference. Enhanced by bottom-side heat dissipation, the device functions effectively across a wide temperature range from 5.77 °C to 109.16 °C and responds rapidly in just 2.03 seconds. Through the self-developed application, each panel pixel on the device can be independently temperature-controlled, allowing for pre-programmed alterations in the shape and color of the concealed target to enable infrared deception. Additionally, a kirigami structure is employed to enhance the device's bendability, facilitating the implementation of curved camouflage and wearable IR information transmission.

Infrared (IR) concealment is essential for reducing the visibility of individuals and vehicles to modern IR surveillance technologies. This involves the ability to diminish or absorb IR radiation, resulting in limited detection and recognition capabilities within the IR spectrum[1–3], particularly in the mid-wave IR (MWIR) (3–5 μm) and long-wave IR (LWIR) (7–14 μm) ranges[4,5].

According to the Stefan-Boltzmann law $M = \varepsilon \sigma T^4$, where $\sigma$ is the Stefan-Boltzmann constant, the radiant heat energy ($M$) is directly proportional to the emissivity ($\varepsilon$) and the fourth power of the

temperature (T) of the object[6]. Traditionally, IR concealment strategies have primarily focused on two approaches. The first one focuses on developing materials with low $\varepsilon$ to reduce IR radiation emissions[3,4,7–11]. The other one relies on employing superstructures and metamaterials to realize thermal shielding[12–14]. For example, vanadium dioxide ($VO_2$), a phase transition metal oxide that undergoes a semiconductor-to-metal transition at 68 °C, has been widely utilized for IR camouflage due to its negative differential thermal emittance across its critical temperature ($T_c$)[9]. To broaden the applicability of $VO_2$, Li et al.[10] used

[1]The Institute of Technological Sciences, Wuhan University, Wuhan, China. [2]Key Laboratory of Artificial Micro-structures of Ministry of Education, School of Physics and Technology, Wuhan University, Wuhan, China. [3]School of Power and Mechanical Engineering, Wuhan University, Wuhan, China. [4]Institute of Physics (IA), RWTH Aachen University, Aachen, Germany. [5]School of Physics and Microelectronics, Key Laboratory of Materials Physics of Ministry of Education, Zhengzhou University, Zhengzhou, China. [6]These authors contributed equally: Yue Hou, Xiaosa Liang, Zhaoyu Li. ✉e-mail: yu@physik.rwth-aachen.de; zywang@whu.edu.cn

W/Al-doped $VO_2$ to achieve a reduced $T_c$ of 43.8 °C. However, phase transition metal oxides typically exhibit a slow response (on the order of minutes), and their IR concealment temperature range is inherently limited by their $T_c$, rendering them unsuitable for applications such as human body IR concealment. On the other hand, conventional thermal shielding approaches utilize wavelength-dependent meta-structures (e.g., polymer microspheres and micro-columnar arrays) that absorb/scatter photons, along with composite metamaterials blocking thermal radiation[15,16]. While effective in narrow bands (e.g., MWIR/LWIR), these methods face inherent bandwidth limitations due to their resonant design constraints, limiting the enablement of general applicability across a broad IR spectrum.

Despite the effectiveness of $\varepsilon$ adjustment and the utilization of thermal insulation materials in adaptive IR concealment technologies, the limited working temperature range ($T_{range}$) significantly constrains the effectiveness of concealed objects in environments with extreme temperatures, e.g., cold or extremely hot. Particularly in sub-ambient and near-ambient temperature scenarios, $\varepsilon$-focused strategies alone cannot achieve stable concealment without active cooling and heating that is enabled by thermoelectric devices (TEDs). To address these limitations, thermoelectric materials, capable of converting electrical power into heating and cooling[17–19], offer a promising solution for active concealment by adjusting surface temperature ($T_s$) to match the ambient temperature ($T_{amb}$). Besides, compared with meta-structures, the thermoelectric temperature-normalization strategy eliminates spectral dependencies by directly regulating surface temperature to match the background. This inherently broadband approach ensures camouflage across all IR bands through physical temperature alignment, circumventing the need for complex multi-band meta-structures.

The pioneering application of thermoelectric materials for IR image display was demonstrated by Venkatasubramanian et al.[20] 2001, where high-performance thin-film superlattices achieved controlled surface temperature patterns. At this stage, research on TEDs for IR concealment remains nascent, and initial studies have primarily focused on demonstrating their feasibility for dynamic IR concealment[20–22]. However, the increasing sophistication of IR detection equipment and the complexity of real-world detection scenarios demand higher requirements for IR concealment systems. Specifically, uniform array driving and display with high color rendering uniformity are crucial, necessitating fast and uniform $T_s$ control from the TED. Furthermore, while thermal camouflage offers concealment, thermal deception provides a more effective means of misleading detection systems. This requires diverse functionalities for advanced IR camouflage systems. However, existing TEDs often suffer from limitations such as susceptibility to heat flux interference from ambient sources (e.g., solar irradiation, nearby heat emitters), which disrupts TED temperature control precision or leads to the presence of distinct color boundaries arising from their longitudinal heat transfer characteristics[23,24].

In this work, we present an IR concealment cloak incorporating active concealment, deception, and messaging functionalities, enabled by an integrated thermoelectric device (IR-TED). To address the challenges aforementioned, several key strategies have been implemented. First, to enhance camouflage performance, a top porous Ethylene-Vinyl Acetate (EVA) layer of 2 mm with low-reflectivity (maximum value of 2.28% in mid- and long-wave) and low thermal conductivity (0.271 and 0.267 W/m K at 313.15 K for transverse/longitudinal heat transfer characteristics) was employed, resulting in a uniform color display with a standard deviation of 0.19 at a driving current of 0.1 A. Second, the bottom circuit was optimized to enable the panel to display graphical information, facilitating both illusionary and messaging functions, while mitigating localized temperature increases due to high driving currents. Furthermore, the integration of heat sinks expanded the operational temperature range $T_{range}$ to 18.72 °C below and

85.28 °C above the $T_{amb}$, which is the widest $T_{range}$ reported to date. Remote circuit control empowers the cloak with powerful deception capabilities, enabling dynamic adjustments to both the temperature and the shape of the concealed object with a rapid response time of 2.03 seconds. Finally, the incorporation of a kirigami structure facilitates stress release, enabling the IR cloak to be easily bent for applications such as curve camouflage, self-camouflage, and wearable information transmission.

## Results

The proposed IR concealment device is designed specifically to conceal objects from IR cameras, not visible light. As shown in Fig. 1a, the proposed IR-TED features enhanced capabilities in both IR deception and messaging. The deception function allows concealed objects to be detected in an alternative form (e.g., different shapes and temperatures) and varied states of motion (e.g., direction of movement), thereby providing a more robust deception effect. Concurrently, the messaging function transmits information back to the observer. Photographs of the front and back sides of the IR-TED are presented in Fig. 1b, c. As shown in Fig. 1d, the core structure of the IR-TED consists of a flexible TED composed of a 6 by 8 array of TE pixels, each of which comprises eight TE pairs housed within cuboids in dimensions of 1.4 mm × 1.4 mm × 2.5 mm (length × width × height). Fabrication details are provided in Figs. S1 and S1 of the Supplementary Information. In Fig. 1e, to control the entire panel, ESP 32 and DRV 8212 chips were utilized for device management and driving, respectively. To prevent the detection of heat radiation by IR cameras, several strategies were implemented to mitigate heat emissions from the control circuit. These strategies include: (1) using a driver chip with lower resistance; (2) designing larger copper coverage for the power supply lines to decrease the Joule heat; and (3) using pulse width modulation signals to apply currents that vary from −100% to +100% for precise temperature regulation. The system achieves rapid thermal response with high precision, demonstrating a minimum resolvable temperature of 0.21 K during cooling and 0.156 K during heating (Fig. S3 in Supplementary Information). Notably, this performance is sustained across an operating power range of 1.14 W to 6.42 W, corresponding to driving currents of 0.38 A to 2.14 A, respectively (detailed power consumption profiles and their estimated battery life are listed in Tables S1 and S2). Circuit implementation strategies enabling this performance are further illustrated in Supplementary Figs. S4–S6.

To ensure a more uniform and durable display with an expanded cooling temperature range, efforts were made to enhance heat-spreading across the device surface and heat dissipation from the bottom. This resulted in superior IR camouflage and display performance. The radar chart in Fig. 1f compares our concealment device against works previously reported across multiple dimensions, including functionality (disguise and information display capabilities), response time, minimum resolvable temperature, and maximum cooling and heating temperature ranges[4,7,21,22,25–27] Compared to previous studies on IR concealment, our device represents an optimal choice in terms of functionality and performance, particularly with a broader temperature range for cooling, thus expanding potential application scenarios. Specific details will be discussed in the following sections.

### Device display performance and temperature threshold optimization

As previously indicated, devices designed for IR camouflage often suffer from narrow operational $T_{range}$ and pronounced boundary lines that can easily be detected, particularly the distinct boundary lines prevalent in numerous reported IR concealment approaches[28–30]. Additionally, traditional top-layer materials tend to have low transmittance and high reflectivity, making them susceptible to the thermal radiation emitted by surrounding objects. To address these

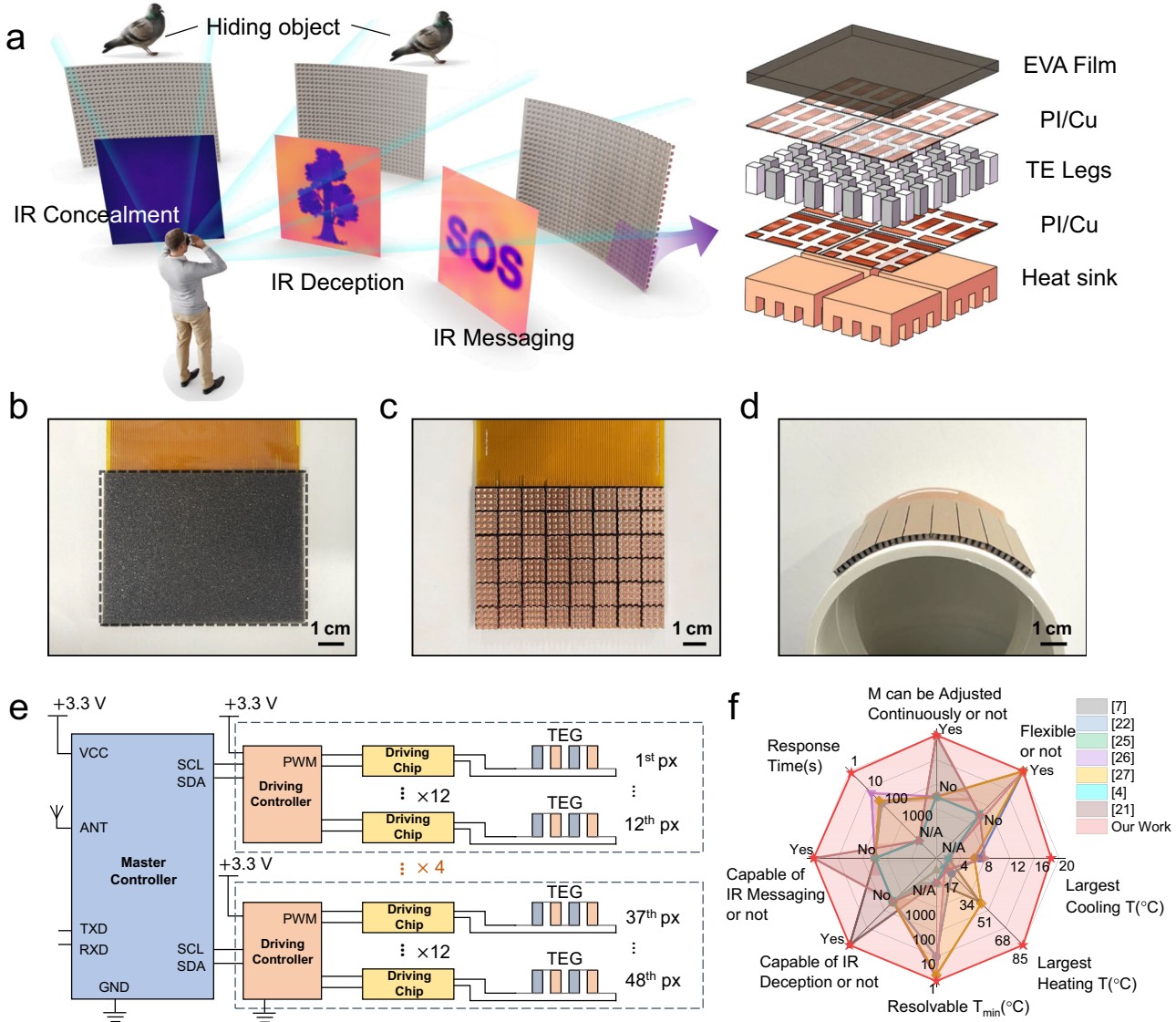

**Fig. 1 | Concept of multifunctional IR-TED. a** Schematic diagram of the IR-TED with three main functions and its specific structural design. **b**, **c** Photographs of the front and back sides of IR-TED. **d** Curve surface conformability for the flexible device under its bending state. **e** Device driving circuit design. **f** Comparison radar map of eight main features between IR-TED and previous IR concealment devices.

challenges, we employed a 2 mm-thick black EVA film as the top anti-reflection and heat-spreading layer with a weight of 0.24 g. Simulations and experiments were conducted to compare the performance of a basic thermal camouflage device (C-TED), a device with a top EVA layer (CE-TED), and a device equipped with both an EVA film and a heat sink (CE-h-TED) (Fig. 2a). As shown in Fig. 2b and S7, the EVA film features a highly porous structure, accounting for its low thermal conductivity of 0.267 W/m K (longitudinal) and 0.271 W/m K (transverse) at 313.15 K.

The porous structure and black surface of the material also enhance its ability to absorb light. As depicted in Fig. 2c, the maximum reflectivity of the upper surface of CE-TED in the mid-IR and long-wave IR bands decreases significantly from 32.57% to a maximum of 3.32% when compared to the C-TED, substantially reducing the influence of environmental thermal radiation. However, experimental results depicted in Fig. 2d indicate that, under identical driving currents, the maximum cooling temperature ($\Delta T$) of the CE-TED, which incorporates the EVA film, decreases from 10.61 K to 6.44 K, thereby affecting the device's performance. To alleviate this issue, a metal heat sink was added to the hot end, resulting in an improvement of 1.16 K in the maximum cooling performance of the CE-h-TED relative to the C-TED.

Notably, the duration of low-temperature maintenance was greatly enhanced; the retention time for a temperature drop of 4.5 K increased from 5 seconds for C-TED and 4 seconds for CE-TED to 23 seconds for CE-h-TED. These optimizations ensure both a uniform temperature-color display and an extended wide-temperature operational range for the IR camouflage device. It is worth noting that while the addition of a heat sink enables the CE-h-TED to achieve prolonged cooling—significantly enhancing the effectiveness of IR camouflage and deception—the linearity of the boundary is less critical for information display purposes. Here, the temperature difference between the display area (low or high temperature) and the non-display area is crucial for effective information transmission, making CE-TED the optimal choice due to its flexibility required for human wearability and anti-interference performance.

The incorporation of a 2 mm EVA film restricts longitudinal heat transfer due to its high thermal resistance, narrowing the operational temperature range. Nevertheless, the increased structural thickness distributes transverse heat flow across additional conduction paths, reducing thermal gradients and homogenizing temperature distribution at display boundaries (Detailed mechanistic analysis is elaborated

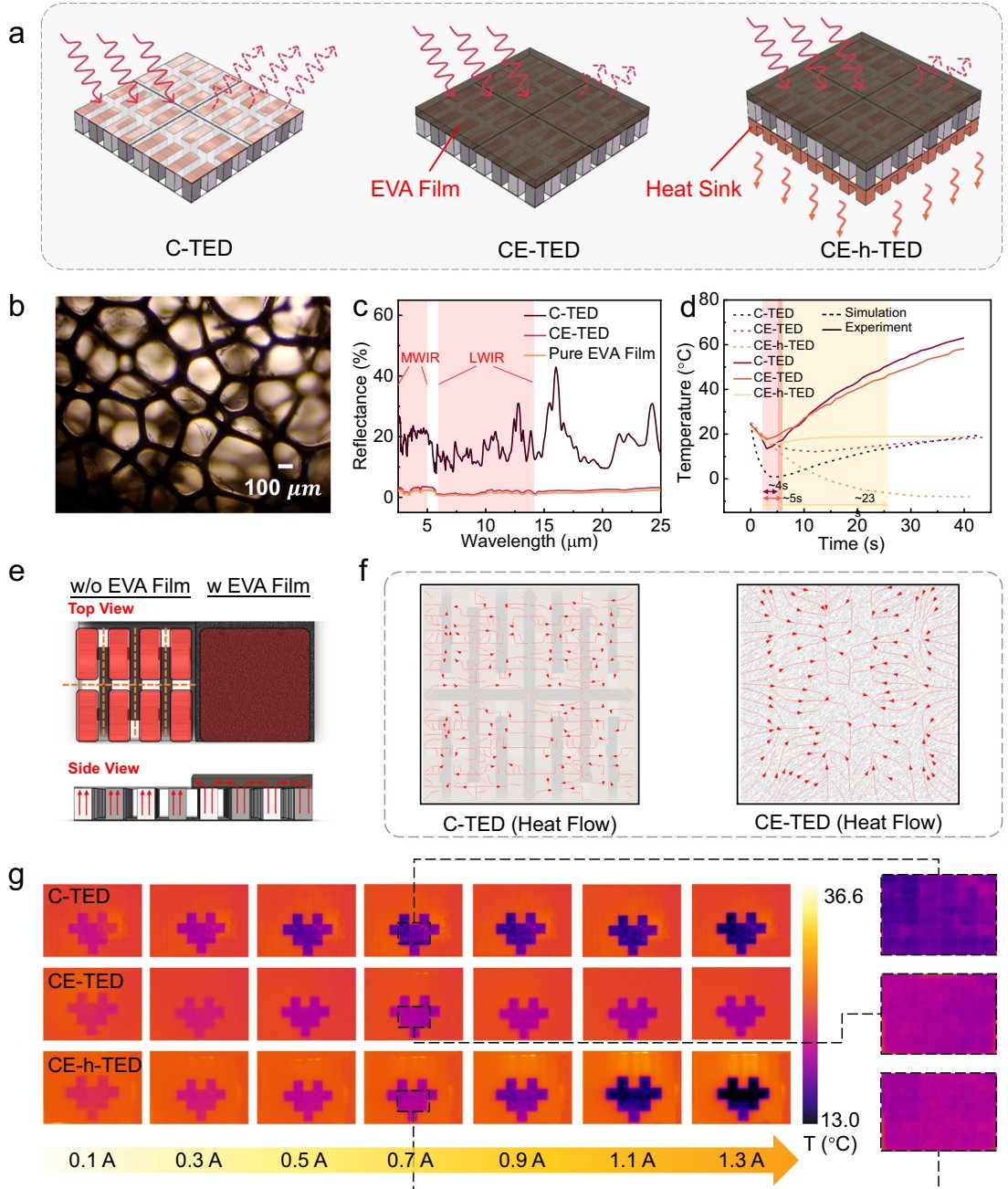

**Fig. 2 | Performance optimization of IR-TED. a** TEDs with different investigated structure designs. **b** The microscopic appearance of the EVA film. **c** Reflectance spectrums for pure EVA film, C-TED, and CE-TED. **d** Simulation and experimental results on the cooling performance of three types of TEDs. **e** Schematic diagram of the comparison between devices with and without the top EVA layer. **f** Finite element simulation results on the heat flow of the upper surface for C-TED and CE-TED. **g** Heart image demonstration for three types of TEDs under different driving currents.

in Figs. S7–1 and S7–2). To further illustrate the uniform heating effect of the EVA film on the top surface of the device, we conducted simulations and experiments to validate the device's performance. Figure 2e shows schematic diagrams of models with and without the EVA film. In the absence of the EVA film, heat conduction occurs from the bottom to the top, resulting in heat accumulation in the electrode regions atop the thermoelectric legs. This creates an uneven temperature distribution on the upper surface, leading to visible temperature gradient boundaries when observed with an IR camera. The simulation of heat flow distribution on the top layers demonstrates that, under consistent contact temperatures at the bottom, the heat flow distribution in the CE-TED with the EVA film is more uniform than in the C-TED (Fig. 2f). It is worth noting that similar thermal

management effects could also be achieved using other porous materials with high thermal resistance, which are expected to facilitate transverse heat-spreading and suppress longitudinal conduction. Further exploration of such materials may broaden the applicability and optimization of thermal homogenization strategies.

Subsequently, we displayed heart-shaped patterns using all three types of devices under different driving currents (Fig. 2g). It should be noted that all photographs were captured at 10 s. As can be seen from Fig. 2g, the C-TED cools faster than the heat sink-equipped CE-h-TED, resulting in the C-TED exhibiting a lower T at 10 s. In contrast, the CE-TED demonstrates less effective cooling than the C-TED due to the additional introduction of EVA thermal resistance material on top of the C-TED structure, which can be well-explained by the heat transfer

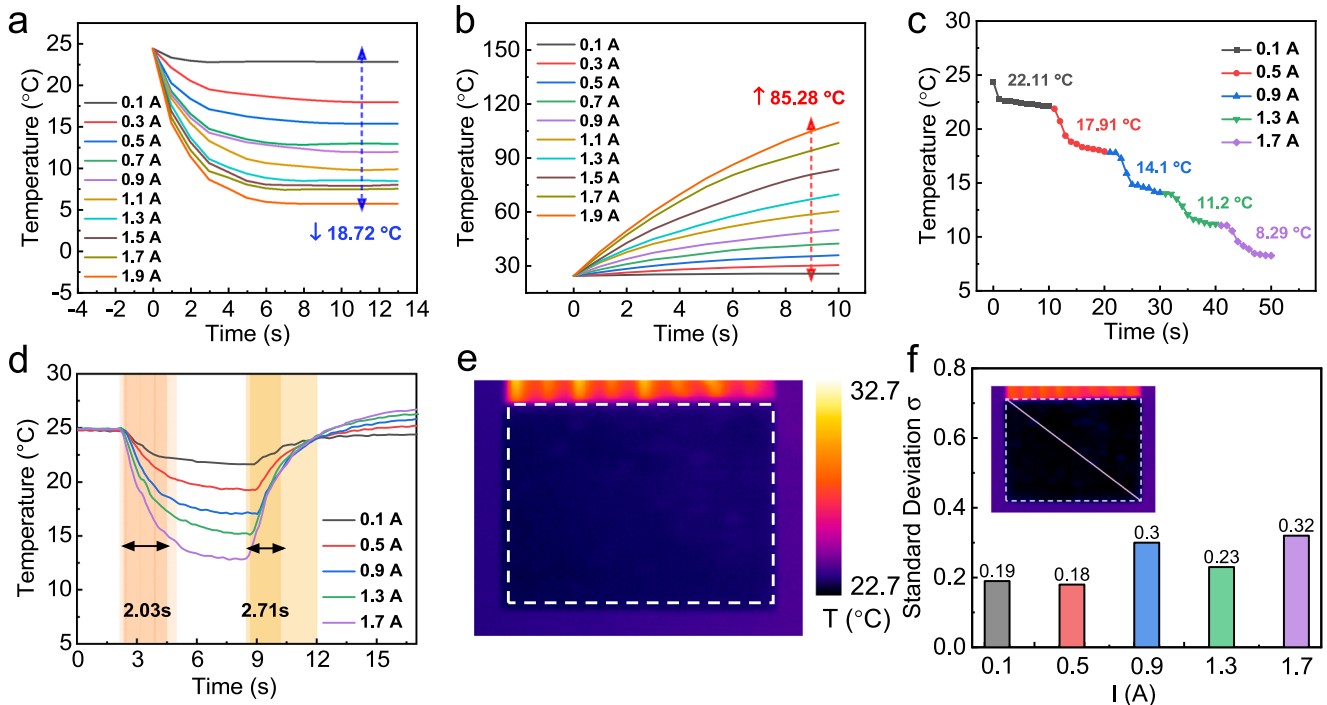

**Fig. 3 | Device performance and thermal display effect of IR-TED. a** Cooling performance of CE-h-TED under the applied current varies from 0.1 A to 1.9 A. **b** The heating performance of CE-h-TED under the applied current varies from 0.1 A to 1.9 A. **c** Continuous cooling performance under the driving current from 0.1 A to 1.7 A. **d** Response and recovery time of CE-h-TEG. **e** Thermal displaying performance for CE- TED under the applied current of 0.1 A. **f** Standard deviation of temperature profiles measured diagonally across the display panel at fixed driving currents from 0.1 A to 1.7 A.

principles (see Figs. S7–1 and S7–2 for details). Under elevated driving currents (>0.9 A), all three devices exhibit accelerated cooling rates, achieving their respective minimum temperatures earlier. Consequently, the CE-h-TED demonstrates the most substantial temperature depression at high currents due to its superior heat dissipation capacity. The magnified view in Fig. 2g demonstrates that the CE-TED and CE-h-TED, equipped with the EVA film, effectively reduce the visibility of temperature gradient boundaries, further validating the uniform heating effect of the EVA material. Device mass measurements (Table S3) reveal distinct weight profiles: C-TED (32.24 g), CE-TED (32.48 g), and CE-h-TED (93.2 g). The EVA film contributes a marginal 0.24 g mass increment, whereas the heat sink introduces a substantial 60.72 g addition. This mass penalty is counterbalanced by the heat sink's critical role in enabling significant thermal display enhancement.

To achieve perfect camouflage, the device must simultaneously adapt to a wide range of ambient temperatures while displaying images with minimal flaws. Here, we evaluated the cooling and heating performance of the CE-h-TED. Based on the Peltier effect of thermoelectric materials[31,32], the temperature of the top layer can be adjusted by reversing the direction of the driving current, thereby enabling both cooling and heating. As shown in Fig. 3a, b, when the driving current was varied from 0.1 A to 1.9 A in 0.2 A increments while maintaining a constant voltage of 3 V, both the cooling and heating temperatures were improved. At a driving current of 1.9 A, the device achieved its maximum cooling and heating performance, reaching a minimum temperature of 5.77 °C and a maximum temperature of 109.76 °C. This flexibility allows the device to cover a broad operational $T_{range}$ of 18.72 °C below and 85.28 °C above the $T_{amb}$, representing the largest $T_{range}$ ever reported to the best of our knowledge. Figure 3c illustrates the cooling temperature variations during continuous changes in driving current (from 0.1 A to 1.7 A with intervals of 0.4 A, with each current maintained for 10 seconds). The results indicate that the CE-h-TED could achieve sustained dynamic temperature regulation, thereby better adapting to complex and

varying environmental conditions. Additionally, we quantified the device's cooling response time using thermocouples for enhanced time resolution (Fig. 3d). With a driving current of 1.7 A, the response time and recovery time were recorded as 2.71 s and 2.03 s, respectively. It is worth noting that such second-level response performance is comparable to state-of-the-art thin-film TEDs, which also typically exhibit cooling responses on the order of several seconds, while the thin-film type potentially offers improved energy efficiency[33]. In comparison, other thermal regulation strategies (e.g., active electrical T control, such as graphene fabric film[26] or passive thermal regulation emissivity-tuning material[27]) typically exhibit response times on the order of minutes, making the second-level response of TEDs more suited to practical applications.

As discussed in the device optimization section, one of the key advantages of the CE-h-TED is its uniform temperature display across the entire panel. Figure 3e shows the cooling effect of the array device operating at a driving current of 0.1 A (demonstrations of higher current input were shown in Fig. S8), demonstrating uniform display performance, which rivals traditionally rigid TEDs driven individually[16,21] Fig. 3f provides a statistical analysis of the mean square deviations of displayed temperatures for various cooling images, indicating that, across driving currents ranging from 0.1 A to 1.7 A with intervals of 0.4 A, the standard deviation ($\sigma$) of temperatures along the image diagonal were 0.19, 0.18, 0.30, 0.23, and 0.32, respectively. These low mean square deviation values suggest that the array device exhibits excellent overall display uniformity, facilitating effective IR concealment and deception.

Additionally, by incorporating a kirigami structure into the FPCB electrodes on the upper layer, concentrated stress at the cut locations is effectively released, thereby improving the bending performance of the device (verified in Fig. S9). The flexibility and stability of the device were verified through long-term bending tests, demonstrating relatively low variation in the randomly selected inner resistance of the C-TED (Fig. S10). These findings indicate both the reliability of the

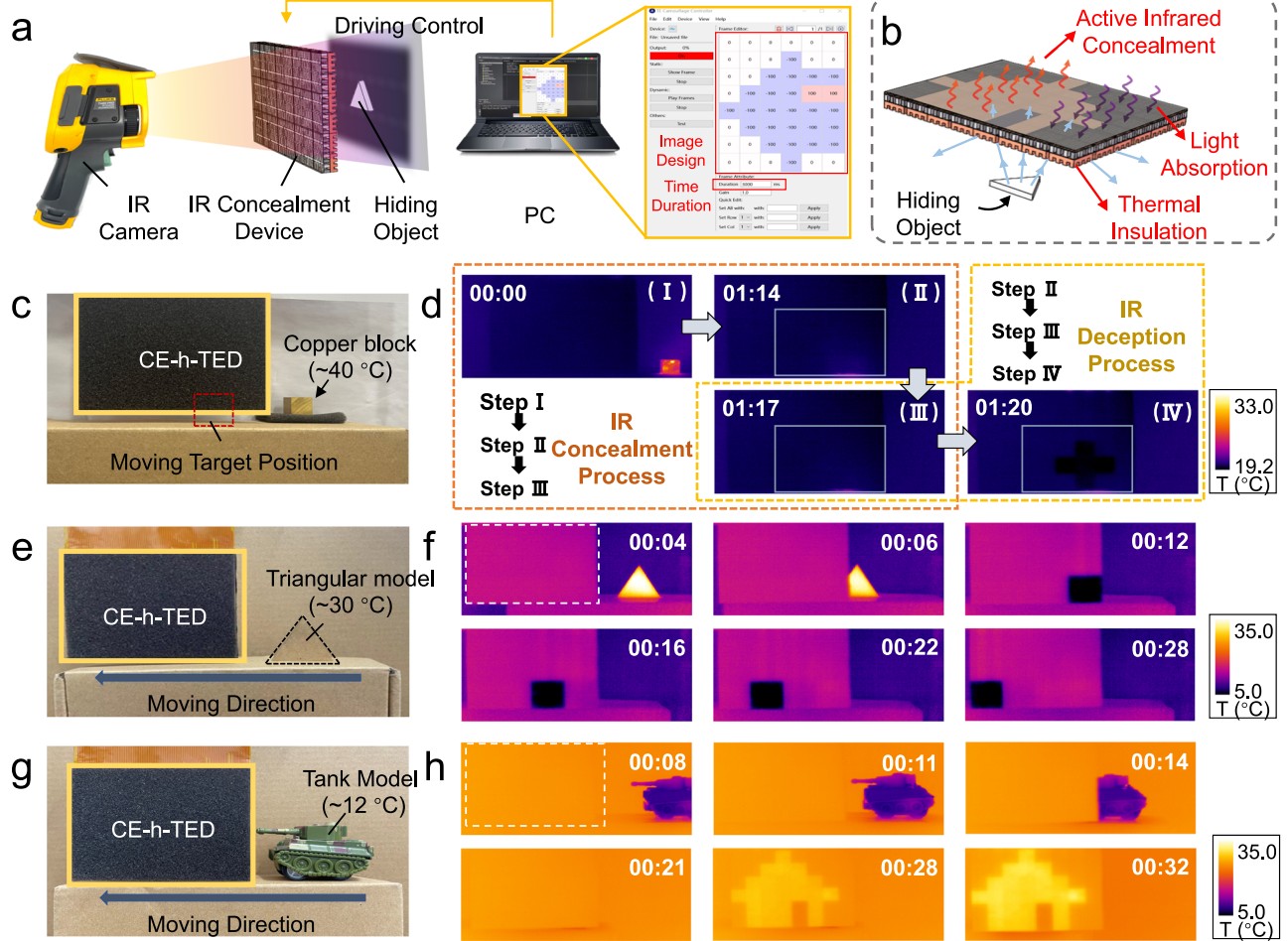

**Fig. 4 | Experiment verification of IR concealment and IR deception function. a** Detection and device driving setup for IR deception function. **b** Schematic diagram of the working principle of the device deception function. **c** Experiment setup for IR concealment and IR deception. **d** The realization of the IR concealment and IR deception. **e** Experiment setup for the 1st experiment for the IR deception function. **f** The realization of the IR deception of a hot triangle model via thermoelectric temperature modulation and shape transformation. **g** Experiment setup for the 2nd experiment for the IR deception function. **h** The realization of the IR deception of a cold tank model via temperature modulation and shape transformation.

device fabrication and the acceptable design of the copper traces in the FPCB.

**Experimental verification of IR concealment and IR deception**

Leveraging the excellent display performance and independently drivable array characteristics of the CE-h-TED device, we implemented IR concealment and deception functionalities through a combination of software development and hardware design. Figure 4a, b illustrates the driving mechanism and working principle of the device. Based on the previously described control circuit and PC-based software, we achieved precise temperature control over 48 individual points on the CE-h-TED surface. This enabled the presentation of predefined images under IR cameras, thereby realizing IR camouflage and deception of the target object. Details of the software development and the testing setup are provided in Figs. S11 and S12 of the Supplementary Information, respectively. Within the CE-h-TED, the metal heat sink layer serves as a thermal insulator, blocking most of the thermal radiation emanating from the target object. However, the complete elimination of thermal radiation from the device itself is not feasible. As illustrated in Fig. 4c, d while attempting to shield a copper block at ~40 °C, thermal radiation gradually reached the display surface after 1 minute and 14 seconds. Upon activating the driving control, the CE-h-TED effectively blocked the thermal radiation, achieving IR concealment within 1 minute and 20 seconds. Subsequently, by adjusting the

positions of the driving array points and varying the driving temperatures, a cross-shaped pattern was successfully displayed on the screen. This deceptive pattern was achieved through simultaneous driving control of both Area 1 and Area 2 (Fig. S13). Specifically, controlling Area 1 enables the concealment of heat radiation from the copper block, while controlling Area 2 facilitates the display of the deceptive pattern

Dynamic processes of the IR deception function were further investigated for objects with temperatures either higher or lower than the $T_{amb}$ (~24 °C). As shown in Fig. 4e, f, when a hot triangular model (~30 °C) entered the region covered by the device, partial thermal camouflage of the hot triangle was observed when the device was inactive, indicating the thermal insulation effect of the device when the model temperature was not excessively high. Simultaneously, as the triangular model enters the back of the panel area, the software was used to precisely set the control current and driving time for each pixel. This enabled the display of a moving cold square along the predetermined path of the triangle, effectively achieving IR deception by dynamically altering both the temperature and shape of the object. Figure 4g, h demonstrates the deception function for a dynamically moving tank model (~12 °C) camouflaged as a static house. These two examples effectively demonstrate the device's capability to successfully camouflage both hot and cold objects and further achieve IR deception through dynamic adjustments in both shape and

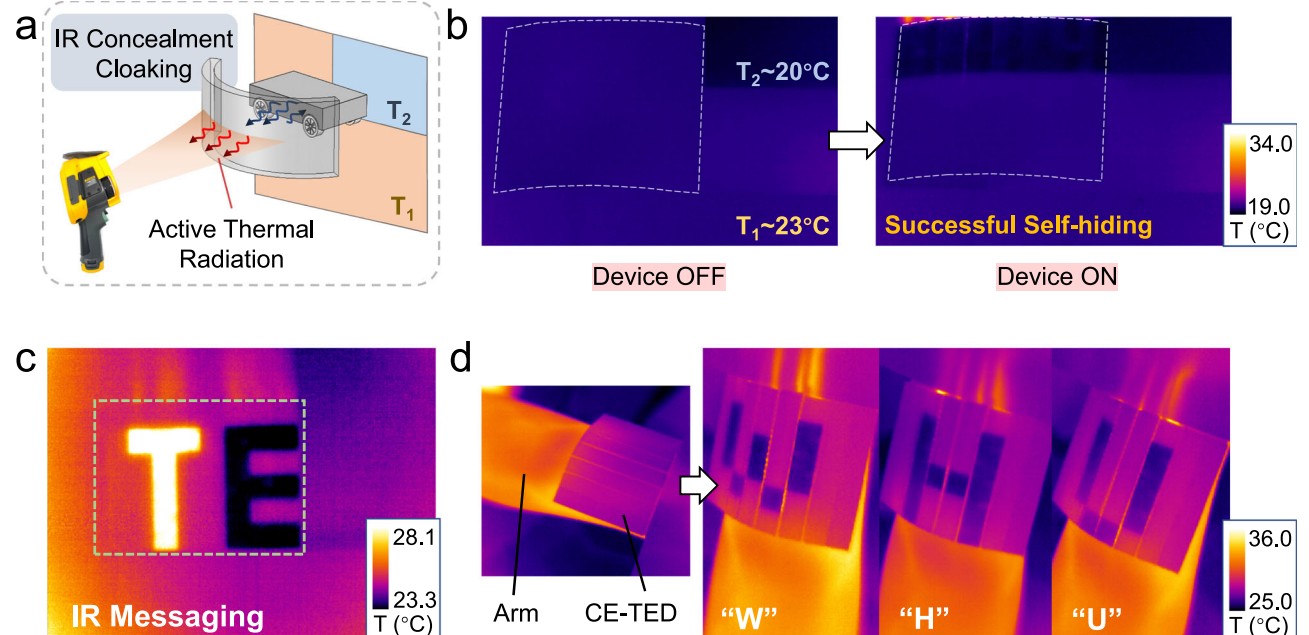

**Fig. 5 | Experiment verification of IR concealment and messaging function for CE-TED. a** Detecting setup for IR concealment for curved CE-TED and the realization of IR concealment for a low-temperature model car in a mixed-temperature environment. **b** Successful self-hiding and object IR concealment for curved CE-TED when the device was on. **c** Demonstration of the IR messaging function for "TE", the abbreviation of thermoelectric, at different temperatures. **d** The message of the capital letters "W", "H", and "U" is shown on the curved CE-TED when worn on the arm.

temperature. Comprehensive supplementary validations confirm exceptional operational resilience, including reliable self-concealment functionality (Fig. S14), stable information integrity across thermal environments (Fig. S15), effective humidity resilience (Fig. S16), uncompromised performance under low-dust conditions (Fig. S17), and robust light-interference resistance (Fig. S18), collectively demonstrating the device's field-deployable robustness.

The flexible CE-TED, in addition to inheriting the IR concealment capabilities of the unbendable CE-h-TED, also exhibits enhanced functionalities, especially in IR concealment and wearable IR messaging. Regarding IR concealment, since the CE-TED lacks a metal heat sink, heat radiation from a nearby heated object can penetrate the device and be captured by the IR camera. However, through localized temperature control, the device successfully concealed the transmitted thermal imaging from the target object behind (an example is shown in Fig. S19 in the Supplementary Information).

As shown in Fig. 5a, we used the CE-TED to wrap a low-temperature car model in a curved configuration. Unlike the planar surface camouflage demonstrated with the CE-h-TED, this experiment involved a more complex background with two distinct temperature zones ($T_1$ ~ 20 °C and $T_2$ ~ 23 °C). Beyond concealing the vehicle model, the CE-TED needed to seamlessly blend into the surrounding environment. As evidenced by the IR images in Fig. 5b, precise temperature regulation across different regions of the array enabled effective camouflage within this complex thermal environment.

Another key function is IR messaging. Through programming on the PC interface, static or dynamic information, including letters or images, can be displayed. As shown in Fig. 5c, the abbreviation letters "TE" were displayed in both high and low-temperature states for the two letters, respectively. For wearable applications, skin surface conformability is crucial. Figure 5d demonstrates the CE-TED's ability to be bent and closely adhered to the skin for displaying preset messages. The IR images clearly depict the display of the English initials "WHU" (Wuhan University) in a low-temperature state. Further wearable examples combining both heating and cooling in a bent state are provided in Figs. S20 and S21 of the Supplementary Information.

Notably, the low-temperature images exhibit a significant contrast with the skin temperature (~32 to ~36 °C), ensuring high visibility and recognizability of the displayed information.

## Discussion

In summary, this study presents active thermal concealment cloaks based on a TED array, exhibiting exceptional IR disguise performance. Through homogeneous thermal diffusion and low-reflectivity facilitated by the top EVA film, the CE-h-TED achieves good display uniformity, eliminating distinct boundary lines between pixels. Unlike conventional IR concealment techniques that rely on emissivity engineering and are inherently limited to specific temperature ranges with response times on the order of minutes, the active thermal concealment device achieves rapid response times (on the order of seconds), broad operational compatibility across both cold and hot environments (spanning 5.77 °C to 109.16 °C), and extended cooling duration enabled by integrated heat sinks. This combination of seamless image display, fast response, and broad temperature range enables the device with multiple functionalities, including IR concealment, deception, and messaging. These functionalities are achieved through thermal blocking, displaying predefined misleading images, and conveying predefined messages via controlled driving circuits, respectively. Future implementations targeting low T operation should incorporate additional anti-condensation strategies. While the kirigami design provides some flexibility, the device's deformability remains constrained. Scaling to larger arrays or higher resolutions faces significant challenges, including intensified thermal crosstalk and heightened drive complexity[34–37]; Future work will therefore focus on two parallel tracks: (1) addressing scalability through architectures design, advanced thermoelectric materials (high ZT), and pixel-level thermal isolation designs to enable practical deployment[38], and (2) innovating thermoelectric materials and structures (e.g., thin-film TED[20]) to develop thinner, more flexible devices—ultimately paving the way for IR concealment cloaks that truly resemble invisibility cloaks. We believe this work holds

revolutionary potential for IR thermal concealment technologies in surveillance and broader societal applications.

## Methods

### Materials

N-type $Bi_2Te_{2.7}Se_{0.3}$ and P-type $Bi_{0.5}Sb_{1.5}Te_3$ ingots (1.4 mm × 1.4 mm × 2.5 mm, L × W × H) were purchased from Wuhan SAGREON Co., and EVA film from Dongguan Henglita Packaging Products Co. was laser-cut to match the CE-TED and CE-h-TED surface dimensions (thickness: 2 mm, area: 60 × 80 mm). The 48 copper heat sinks (9 × 9 mm base with 16 fins of 2 mm height) for CE-h-TED were provided by Chongqing Fengreng Feifei Electronic Technology Co., and 8 aluminum alloy heat sinks, along with one driving PCB heat sink (20 mm × 15 mm × 5 mm, L × W × H) were brought from Shenzhen Maoyuan Heat Dissipation Technology.

### Device fabrication section

The PCB and FPCB were all self-designed, and then printed via the Shenzhen Jialichuang Technology Group Co., Ltd. Low-temperature lead-free solder paste with a melting point of 138 °C was purchased from Senju Metal Industry Co., Ltd. During the soldering process, the temperature was set to 180 °C, and the soldering time was around 3 mins. On the back side of the device, 48 heat sinks were contacted by the silicone adhesive with high thermal conductivity. (Detailed fabrication methods are shown in Figs. S1 and S2 in the Supplementary information)

### Materials characterization and device performance measurement

The device flexibility and stability results were tested by the flexible electronic tester (FT2000) from the Prtronic™. The thermal conductivity of the EVA film was measured by the Laser thermal conductivity meter (LFA467) from NETZSCH. All IR images were captured by an IR camera (Ti489 PRO FLUKE). To test the maximum cooling and heating temperature, the adjustable DC power source (SS-3020KDS from A-BF™) was used as the main power supply, and different charging current was controlled via the control interface on PC, which was explained in detail in Supplementary Information. The temperature on the display was recorded by an IR camera. Surface temperatures of C-TED, CE-TED, and CE-h-TED were also measured via thermocouple (K-type, 1 mm diameter, ±0.5 °C accuracy), as shown in Supplementary information (Fig. S22). The device and material reflectivity were tested by the Fourier transform IR spectrometer (Nicolet IS50). All experiments were strictly conducted via natural convection without auxiliary fans.

## Data availability

The source data generated in this study are provided in the Source Data file. Source data are provided with this paper.

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

## Acknowledgements
The authors would like to thank Prof. Zhengyou Liu for his assistance with the COMSOL software. The authors greatly appreciate the financial support from the National Key R&D Program (Grant No. 2023YFB4603800) and the National Natural Science Foundation of China (Grant Nos. 12474093, 12302220).

## Author contributions
Y.Y. and Z.W. supervised the whole project. Y.H. and Z.W. conceived and designed the research. Y.H. and Z. Li conducted research. X.L. and Z.Z. contributed to the software design. Q.D., X.S., C.L., W.C. and W.Z. contributed to the electrical measurement and COMSOL simulation. Y.H., Z.L. and Y.Y. wrote the manuscript and all authors commented on the final draft.

## Funding

## Competing interests
The authors declare no competing interests.
