## [Transparent Peer Review file · Nature Communications]

Thermoelectric Device for Active Thermal Concealment, Deception and Messaging

Corresponding Author: Dr Yuan Yu

Version 0:

Reviewer comments:

Reviewer #1

(Remarks to the Author)

The overall results of the paper are interesting and appears to be a coherent advancement in the application of thermoelectric technology for thermal emission control of surfaces. The arguments presented for the technology, relative to meta surfaces and phase transition metal oxides, are worthy of making this approach aware to the readership of Nature Communications.

But, there are some aspects of the work that need to be clarified and expanded on, before it can be recommended for publication. I have made some comments here for authors consideration.

I thank the authors for continuous numbering of lines from page to page; good idea! Makes the review easier and to give specific feedback.

The idea of using high-speed thermoelectric devices, particularly based on thin-film thermoelectric devices, for IR-emission-based patterns is not new. Please see Fig. 5a (NATURE |VOL 413 | 11 OCTOBER 2001 |www.nature.com) - could be added in their description of background - so that the readership is aware of prior work and also provides a reference when the authors cite thin film TE (line 353).

1) The acronym EVA was not defined, in the abstract, and then it carried forward. Please define and also add some details about EVA in the Methods section.

2) Line 58-59 - the authors mention the complexity of meta structures - it's worth commenting on whether the meta structures approach is intrinsically broadband - as an advantage for using the thermoelectric approach of temperature normalization with respect to the background; this will be good for the readership.

3) Line 63-64 - The IR emission and absorption are equivalent - good absorbers are good emitters - so, I am confused by $T_{amb} < T_{target}$ - needs some clarification on what the authors are purporting to mean.

4) Line 76-77 - the authors need to clarify "existing TEDs suffer from limitations such as susceptibility to external radiation" - it is not clear

5) Line 83-84 - description about EVA layer needed; some data are presented but seems incomplete

6) Line 103-104 - interesting objective - but it was not clear if such a result was clearly demonstrated in this paper. If not, please remove reference to Harry Potter!

7) Line 123-124 - supplementary data not in order; goes from S1, S2 to S11.

8) In Fig. 2g - C-TED is coolest at 0.5 Amp; I would have expected CE-h-TED to be the coolest; can the authors explain why so?

9) Similarly in Fig. 2g at 1.3 Amp - C-TED appears cooler than CE-TED. Why so? Understand why CE-h-TED is the coolest.

- 10) Basically, the relative value of EVA vs Heat sink (HS) is not clear from data; please provide in the supplementary section.
- 11) Line 153-154 - I like the concept of EVA as AR-coat and heat-spreading material in plane; please provide thermal conductivity data for EVA in the plane of the film in the supplementary section.
- 12) Line 173-174 - the description of retention time is interesting - retention is better with heat-sink, understandably, and that means that set of data are different from (9) above; looks like we need Heat sink for temporal stability. This should be discussed in the summary/discussion. Also, did they use a fan in addition to the heat sink - this should be clearly noted; how was the heat dissipated from the heat sink?
- 13) Fig. 3e - is the applied current 0.1 A? or is it much higher? Do we have similar displays for higher currents.
- 14) Line 214-215 - at 1.9 A, $T_c = 5.77\text{C}$, $T_h = 109.8\text{C}$; $T_{amb} = 25\text{C}$. With the ambient being 25C and T_{cold} being significantly low at 5.77C - how was the condensation of atmospheric moisture avoided in the data? Please include these details in the Methods section.
- 15) Line 225-227 - the authors mention passive cooling - it is not clear what they refer to.
- 16) Line 360-361 - the Methods section is very terse; need more description - including covering many of the above points
- 17) Line 351-352 - I was looking to see more data with their Kirigami approach and how it was implemented. Supplementary S1 and S2 do not appear sufficient. Basically, it would be good to know how "flexibility" is achieved
- 18) While IR-imaging data is the most relevant, given the targeted application and theme of the paper, some additional quantitative cooling data for the three cases (C-TED, CE-TED and CE-h-TED) would be good to show that - in all cases, the images are due to emissivity control indeed and not from any actual temperature differences; some limited thermocouple measurements on control test structures for each of the three cases would be convincing.
- 19) Finally the aspect of how much power and energy are used for the TED-based IR-image control, along with weight of TE modules and heat-sink, should be described in the main paper and with some specific data perhaps in the supplementary section - this will be important when readers look at the TED approach to competing approaches using meta-surfaces and phase change materials.

Reviewer #2

(Remarks to the Author)

This manuscript introduces a multifunctional infrared (IR) stealth cloak built from a thermoelectric device (TED) array capable of active camouflage, deception, and messaging. Leveraging fast, programmable heating and cooling via the Peltier effect and integrated circuit control, the authors achieve dynamic temperature shaping with remarkable uniformity, rapid response (~ 2 seconds), and wide operation temperatures ($5.77\text{ }^\circ\text{C}$ to $109.16\text{ }^\circ\text{C}$). Key innovations include the use of a porous EVA film to enhance temperature uniformity and a kirigami-structured flexible PCB enabling wearable applications. The work represents a substantial advancement in the field of active thermal stealth and adaptive IR display technologies.

The main strengths of the presented work include an operational temperature range that is broader than previously reported systems of its kind and a response time (~ 2 seconds) that is significantly faster than conventional materials like VO₂-based systems, which operate over minutes. Moreover, the programmable resolution with low-temperature standard deviation ($\sim 0.19\text{ K}$) per pixel demonstrates very good thermal uniformity. The use of porous black EVA film reduces surface reflectivity from $\sim 32\%$ to $\sim 3\%$, significantly improving stealth performance.

The authors also utilize clever circuit design (PWM, low-resistance driver chips, copper coverage) to minimize parasitic heating and support spatially resolved thermal control and incorporate kirigami patterns to improve mechanical adaptability without sacrificing function, enabling wearable applications.

In general, the quality of the manuscript and the level of the shown results are consistent with the publication in Nature Communications.

However, some comments should be addressed and discussed:

- High functionality is achieved, but it is unclear if the power demands, up to 1.9 A per panel at 3V, could limit battery-powered, portable, or long-duration applications. The authors should provide a quantitative discussion of energy usage over time and thermal-to-electrical efficiency.

- The device uses a 6×8 array (48 pixels), with each pixel consisting of 8 thermoelectric pairs. While effective for proof-of-concept, larger arrays or finer resolution for higher fidelity deception or messaging would likely require significant increases in complexity and power. Some discussion on scaling strategies would be helpful.

- While the kirigami structure allows bending and flexibility, long-term mechanical fatigue, delamination, and thermal cycling stability were not explored in detail. Did the authors test fatigue (e.g., $>1,000$ bend cycles) and resistance to environmental

conditions (e.g., humidity, dust)?

- How would the proposed system perform under varying ambient radiation (e.g., sunlight, rain) or dynamic backgrounds (changing temperature, clouds)? This would be relevant for the eventually sought applications.

- Same as above for wearable applications: a person engaged in intense physical activity could modify the heating of the body. Could that affect the performance of the display?

- Has the approach been tested with adversarial detection models to quantify how well the system can evade modern IR tracking or recognition software? Is there any operation metric (beyond what is reported in Fig. 1f) that allows us to establish how well this system can perform in practice? This would give objective insight into effectiveness beyond controlled lab conditions.

Other minor comments:

- I could not see where EVA is defined.

- The analogy to Harry Potter's invisibility cloak is engaging but should be used with care in formal communication to avoid undermining the technical rigor.

Reviewer #3

(Remarks to the Author)

Version 1:

Reviewer comments:

Reviewer #1

(Remarks to the Author)

Thanks again to the authors for providing a continuously numbered revised text for an easier review as well as for addressing my suggested edits/clarifications.

Here are some suggested edits:

- 1) Line 58-59: instead of saying "effective in specific bands" change to "effective in narrow bands" as that's the focus of that section. Specificity is not the problem; it's the narrowness of the band, that's being addressed.
- 2) Line 59-60: could be changed to "due to their resonant design constraints limiting the enablement of general applicability across a broad IR spectrum".
- 3) Line 64-66: Low-temperature scenarios can mean many things and the authors need to be more specific. Could they say, "Particularly in near-ambient scenarios, ε-focused strategies alone cannot achieve stable stealth without active cooling and/or heating that is enable by thermoelectric devices".
- 4) Line 115-116 - thanks for the clarification and sticking with scientific terms, and avoiding reference to Harry Potter.
- 5) It is good that the authors have revised the supplementary text, with figures and tables being numbered as they appear first in the main manuscript.
- 6) Table R1 and Fig. S7a are understandable and sufficient for this paper, but I suggest that would be good for the authors to note that this EVA usefulness is to be studied further and other materials could be considered for such purposes. Note - this is only a suggestion.
- 7) Thanks for the clarification of the non-use of auxiliary fans.
- 8) Line 259-264: the authors note the speed of cooling with their bulk TECs in seconds, relative to other thermal regulation strategies that take minutes. It is worth noting here, for the general readers, that thin-film thermoelectric devices can be much faster and also energy efficient than bulk TECs (see for example -Fig. 2g, Fig. 2h, Fig. 2i in reference <https://doi.org/10.1038/s41551-023-01070-w>); this also applies to Line 384-385. It would be worth for the readership to know what has been shown/published already.
- 9) Line 397-398: pixel-type thermal isolation and IR imaging has been shown with thin-film thermoelectric devices (see <https://www.nature.com/articles/s41467-025-59698-y>) over a large area. It would be worth for the readership to know what has been shown/published already.

The authors have done very interesting work and explored an important area in the IR emission control of surfaces and have described a variety of possibilities. Hence, I recommend the publication with the above suggested edits and clarifications.

Reviewer #2

(Remarks to the Author)

The authors satisfactorily addressed our comments. I particularly appreciated the experiments at different light intensities. It looks like typical daylight irradiation could hinder the proposed approach but, still, while further work will be needed to extend the operability of the proposed strategies, the shown advancement are relevant. I can recommend the publication.

Version 2:

Reviewer comments:

Reviewer #1

(Remarks to the Author)

Dear Authors,

The edits to the paper look fine.

Regards!

Authors' response to the reviewers

Reviewer #1:

The overall results of the paper are interesting and appears to be a coherent advancement in the application of thermoelectric technology for thermal emission control of surfaces. The arguments presented for the technology, relative to meta surfaces and phase transition metal oxides, are worthy of making this approach aware to the readership of Nature Communications.

But, there are some aspects of the work that need to be clarified and expanded on, before it can be recommended for publication. I have made some comments here for authors consideration.

I thank the authors for continuous numbering of lines from page to page; good idea! Makes the review easier and to give specific feedback.

The idea of using high-speed thermoelectric devices, particularly based on thin-film thermoelectric devices, for IR-emission-based patterns is not new. Please see Fig. 5a (NATURE |VOL 413 | 11 OCTOBER 2001 |www.nature.com) - could be added in their description of background - so that the readership is aware of prior work and also provides a reference when the authors cite thin film TE (line 353).

Response:

Thank you for your thoughtful assessment of our manuscript and your constructive suggestions. We sincerely appreciate your recognition of the paper's coherence and its potential contribution to thermoelectric technology for thermal emission control.

We have carefully considered your comment regarding prior work on thin-film thermoelectric devices for IR-emission-based patterns. As suggested, we have added the **new reference 20** (Nature **413**, 597–602 (2001)) in the background and conclusion sections to acknowledge the pioneering work of IR-emission-based display.

Revisions to the Introduction and Conclusion are shown below.

On page 3: “The pioneering application of thermoelectric materials for infrared image display was demonstrated by Venkatasubramanian *et al.* in 2001, where high-performance thin-film superlattices achieved controlled surface temperature patterns²⁰. At this stage, research on thermoelectric devices (TEDs) for infrared stealth remains nascent, and initial studies have primarily focused on demonstrating their feasibility for dynamic IR stealth²⁰⁻²².”

On page 16: “Future work will therefore focus on two parallel tracks: (1) addressing

scalability through architectures design, advanced thermoelectric materials (high ZT), and pixel-level thermal isolation designs to enable practical deployment, and (2) innovating thermoelectric materials and structures (e.g., thin-film TED²⁰) to develop thinner, more flexible devices—ultimately paving the way for infrared stealth cloaks that truly resemble invisibility cloaks. We believe this work holds revolutionary potential for IR thermal stealth technologies in surveillance and broader societal applications.”

- **Added Reference**

20. Venkatasubramanian, R., Siivola, E., Colpitts, T. *et al.* Thin-film thermoelectric devices with high room-temperature figures of merit. *Nature* 413, 597–602 (2001).

1) *The acronym EVA was not defined, in the abstract, and then it carried forward. Please define and also add some details about EVA in the Methods section.*

Response:

We appreciate the reviewer’s careful observation regarding the undefined acronym. We have now defined EVA (Ethylene-Vinyl Acetate) at its first presence in the **Abstract** and again at the first mention in the **main text** (page 3). Additionally, we have expanded the description of the EVA layer in the manuscript (thermal conductivity and physical dimension) and the purchase information of EVA in the **Methods section**, as shown below:

- **Revision**

On page 1: “By optimizing the circuit design and incorporating a low-reflectivity black porous Ethylene-Vinyl Acetate (EVA) film, ……”

On page 8: “As shown in Figure 2b and S7, the EVA film features a highly porous structure, accounting for its low thermal conductivity of 0.267 W/m·K (longitudinal) and 0.271 W/m·K (transverse) at 313.15 K.”

On pages 16-17: “Materials. N-type Bi₂Te_{2.7}Se_{0.3} and P-type Bi_{0.5}Sb_{1.5}Te₃ ingots (1.4 mm × 1.4 mm × 2.5 mm, L × W × H) were purchased from Wuhan SAGREON Co., and EVA film from Dongguan Henglita Packaging Products Co. was laser-cut to match the CE-TED and CE-h-TED surface dimensions (thickness: 2 mm, area: 60 × 80 mm). The 48 copper heat sinks (9 × 9 mm base with 16 fins of 2 mm height) for CE-h-TED were provided by Chongqing Fengreng Feifei Electronic Technology Co., and 8 aluminum alloy heat sinks along with one driving PCB heat sink (20 mm × 15 mm × 5 mm, L × W × H) were brought from Shenzhen Maoyuan Heat Dissipation Technology.”

On page 17: “Materials characterization and device performance

measurement. ...The thermal conductivity of the EVA film was measured by the Laser thermal conductivity meter (LFA467) from NETZSCH. ...”

2) *Line 58-59 - the authors mention the complexity of meta structures - it's worth commenting on whether the meta structures approach is intrinsically broadband - as an advantage for using the thermoelectric approach of temperature normalization with respect to the background; this will be good for the readership.*

Response:

We appreciate the reviewer's insightful comment regarding the broadband capability of meta-structures. As rightly noted, conventional thermal shielding meta-structures—such as polymer microspheres, micro-columnar arrays, and composite metamaterials—typically exhibit inherent bandwidth limitations. Their photon-absorption and scattering mechanisms rely on resonant structures or tailored emissivity/reflectivity, which are often optimized for specific IR bands (e.g., MWIR or LWIR). Consequently, their effectiveness diminishes outside these targeted wavelengths, necessitating complex multi-band designs to achieve broad spectral coverage.

In contrast, our strategy offers an intrinsic broadband advantage. By actively regulating the object’s surface temperature to match the background via the Peltier effect, the emitted blackbody radiation inherently aligns with the ambient thermal profile across all IR bands. This fundamental temperature-driven approach requires no wavelength-selective structures, enabling seamless camouflage in dynamic environments where background temperatures vary or multi-spectral IR detection is employed. Thus, the TE method provides a universally adaptable solution for broadband IR suppression, complementing and extending beyond the capabilities of meta-structure-based designs.

Revisions to the Introduction content are shown below.

● **Revision**

On page 2: ".....conventional thermal shielding approaches utilize wavelength-dependent meta-structures (e.g., polymer microspheres and micro-columnar arrays) that absorb/scatter photons, along with composite metamaterials blocking thermal radiation^{15,16}. While effective in specific bands (e.g., MWIR/LWIR), these methods face inherent bandwidth limitations due to their resonant design constraints.”

On page 3: “....., Besides, compared with meta-structures, the thermoelectric temperature-normalization strategy eliminates spectral dependencies by directly regulating surface temperature to match the background. This inherently broadband approach ensures camouflage across all IR bands through physical temperature

alignment, circumventing the need for complex multi-band meta-structures.”

3) *Line 63-64 - The IR emission and absorption are equivalent - good absorbers are good emitters - so, I am confused by $T_{amb} < T_{target}$ - needs some clarification on what the authors are purporting to mean.*

Response:

We thank the reviewer for this insightful clarification regarding Kirchhoff's law of thermal radiation. The original statement in Lines 63-64 (" ϵ adjustment is only feasible when the target surface temperature (T_s) exceeds the ambient temperature (T_{amb}).") was an oversimplification that neglected the fundamental equivalence of emission and absorption.

Although emissivity (ϵ) modulation is governed by Kirchhoff's law of thermal radiation ($\alpha = \epsilon$ for any wavelength), its efficacy for IR stealth critically depends on the target-to-ambient temperature gradient. When $T_s < T_{amb}$, objects become net absorbers of ambient IR radiation, even with low ϵ , causing surface heating that compromises thermal concealment.

Besides, to achieve efficient thermal stealth, the radiant heat energy (M) of the object should be equivalent to that of its surroundings. Here, the radiant heat energy (M) is directly proportional to the emissivity (ϵ) and the fourth power of the temperature (T) of the object ($M = \epsilon\sigma T^4$). When the object is cooler than the environment, i.e., $T_s < T_{amb}$, the emissivity of the object must be significantly increased to counterbalance its lower temperature for achieving comparable radiant heat energy. This is very challenging since the ϵ value is limited to a maximum of 1.0, while the temperature term shows a much stronger effect. Thus, ϵ -focused strategies alone cannot achieve stable stealth in sub-ambient conditions without active cooling.

We have revised this section to explicitly acknowledge that emissivity modulation remains physically possible regardless of temperature differentials (per Kirchhoff's law), while emphasizing its practical limitation when $T_s < T_{amb}$. Revisions to this section are shown below.

● **Revision**

On pages 2-3: "...Particularly in low-temperature scenarios, ϵ -focused strategies alone cannot achieve stable stealth without active cooling"

4) *Line 76-77 - the authors need to clarify "existing TEDs suffer from limitations such as susceptibility to external radiation" - it is not clear*

Response:

We appreciate the reviewer's request for clarification regarding the limitations of existing TEDs. The phrase "susceptibility to external radiation" in the main text refers specifically to heat flux interference from ambient sources (e.g., solar irradiation, nearby heat emitters) that disrupts TED temperature control precision. This occurs because TED panels do not have the radiation shielding layers. Thus, we have added the black porous structured EVA with low reflectivity to avoid this interference. **We have revised the text to explicitly state.**

● Revision

On page 3: "...However, existing TEDs often suffer from limitations such as susceptibility to heat flux interference from ambient sources (e.g., solar irradiation, nearby heat emitters) that disrupt TED temperature control precision or lead to the presence of distinct color boundaries arising from their longitudinal heat transfer characteristics^{23,24}."

5) Line 83-84 - description about EVA layer needed; some data are presented but seems incomplete

Response:

We sincerely appreciate the reviewer's valuable feedback regarding the incomplete description of the EVA layer. We have revised the relevant section to include critical thermal and optical properties of Ethylene-Vinyl Acetate (EVA). **Revisions to the main text are shown below.**

● Revision

On page 3: "...First, to enhance camouflage performance, a top porous Ethylene-Vinyl Acetate (EVA) layer of 2 mm with low reflectivity (maximum value of 2.28% in mid- and long-wave) and low thermal conductivity (0.271 and 0.267 W/m·K at 313.15 K for transverse/longitudinal heat transfer characteristics) was employed, ..."

6) Line 103-104 - interesting objective - but it was not clear if such a result was clearly demonstrated in this paper. If not, please remove reference to Harry Potter!

Response:

We sincerely thank the reviewer for raising this valid concern. As shown in Fig. 4, our device achieves infrared stealth by obscuring objects from thermal detection—a fundamentally distinct mechanism from the fictional visible-light invisibility portrayed in *Harry Potter*. Acknowledging the need for formal communication, we have removed

the cultural reference and revised the text to:

- **Revision**

On page 5: "The proposed IR stealth device is designed specifically to conceal objects from infrared cameras, not visible light."

7) Line 123-124 - supplementary data not in order; goes from S1, S2 to S11.

Response:

We sincerely appreciate your careful observation of the supplementary figure organization. Upon completing all revisions and incorporating new content (environmental resistance tests, etc.), we have comprehensively reorganized the Supplementary Information. All figures and tables are now sequentially numbered according to their first appearance in the text (Figs. S1-S22, Tables S1-S3), with cross-checked in-text citations to ensure consistency.

8) In Fig. 2g - C-TED is coolest at 0.5 Amp; I would have expected CE-h-TED to be the coolest; can the authors explain why so?

Fig. R1 Equivalent longitudinal thermal resistance schematics of single legs for three thermoelectric devices: **a.** C-TED, **b.** CE-TED, and **c.** CE-h-TED.

Response:

We thank the reviewer for this insightful observation. The thermal image in Fig. 2g was captured at 10 seconds during cooldown. We deliberately chose this time to ensure fair device comparison, as devices without heat sinks exhibit temperature rebound in later stages due to inadequate heat dissipation, yet in the initial phase, C-TED cools

faster than heat sink-equipped CE-h-TED. This explains why C-TED appears cooler at 0.5A in Fig. 2g. Based on the heat transfer theory, we analyzed the reasons for the higher cooling speed of C-TED during the initial cooling stage below.

Starting from the heat balance equation at the cold side:

$$C_c dT_c/dt = -(Q_p - Q_k - Q_{j,c}) \quad (\text{Eq.1})$$

Where C_c is the effective thermal capacity at the cold side (including the thermal capacity of the layered structure near the cold side), $Q_p = \alpha IT_c$ is the Peltier cooling power (heat absorption), $Q_k = G(T_h - T_c)$ is the heat leak from the hot side to the cold side due to thermal conduction (G is the thermal conductance from the cold side to the hot side), and $Q_{j,c}$ is the contribution of Joule heating to the cold side; typically, Joule heating is distributed evenly along the thermoelectric leg, with about half affecting the cold side, i.e., $Q_{j,c} \approx \frac{1}{2} I^2 R$.

At the initial cooling stage ($t=0+$), $T_c = T_h = T_0$ (ambient temperature), so $Q_k = 0$. Equation 1 simplifies to:

$$C_c dT_c/dt = -(\alpha IT_0 - \frac{1}{2} I^2 R) \quad (\text{Eq.2})$$

Then the cooling rate can be solved as:

$$\left. \frac{dT_c}{dt} \right|_{t=0} = -\frac{\alpha IT_0 - \frac{1}{2} I^2 R}{C_c} \quad (\text{Eq.3})$$

The negative sign indicates a temperature drop. If the net cooling power is positive, the initial cooling rate is inversely proportional to C_c :

$$\left. \frac{dT_c}{dt} \right|_{t=0} \propto \frac{1}{C_c} \quad (\text{Eq.4})$$

This shows that a smaller cold-side thermal capacity results in a faster initial cooling rate. This is the core reason why C-TED shows the fastest initial cooling speed (Effective cold-side thermal capacity C_c of the three devices: $C_{c, C-TED} < C_{c, CE-TED} = C_{c, CE-h-TED}$). **For effective cold-side thermal capacity C_c** , as shown in Fig. R1, adding an EVA porous membrane layer (low thermal conductivity) increases the cold-side thermal capacity $C_{c, CE-TED} = C_{c, C-TED} + C_{EVA} > C_{c, C-TED}$. For CE-h-TED, the hot-side heat sink does not affect the cold side, so $C_{c, CE-h-TED} = C_{c, CE-TED}$. Therefore, $C_{c, C-TED} < C_{c, CE-TED} = C_{c, CE-h-TED}$.

For the subsequent process ($t > 0$), the temperature change follows an exponential decay with time constant $\tau = C/G$ (C is the total system thermal capacity, approximated as cold-side thermal capacity C_c ; G is the total internal thermal conductance). Assuming a small temperature difference, solving the heat equation yields an exponential form:

$$T_c(t) = T_0 - \Delta T_{\max}(1 - e^{-t/\tau}) \quad (\text{Eq.5})$$

The general cooling rate expression is:

$$\frac{dT_c}{dt} = -\frac{\Delta T_{\max}}{\tau} e^{-t/\tau} \quad (\text{Eq.6})$$

Where ΔT_{\max} is the maximum possible temperature difference. Thus,

$$\left. \frac{dT_c}{dt} \right|_{t=0} \propto \frac{G}{C_c} \quad (\text{Eq.7})$$

For internal thermal conductance G , the EVA porous membrane layer has low thermal conductivity, increasing the thermal resistance from the cold side to the thermoelectric leg, thus reducing thermal conductance: $G_{\text{CE-TED}} < G_{\text{C-TED}}$. For CE-h-TED, the hot-side heat sink only affects the thermal resistance from the hot side to the environment, not the internal thermal conductance (from cold side to hot side), so $G_{\text{CE-h-TED}} = G_{\text{CE-TED}}$. Thus, $G_{\text{C-TED}} > G_{\text{CE-TED}} = G_{\text{CE-h-TED}}$.

Then, coming back to Eq.7, we can conclude that device C-TED has the fastest cooling rate in the initial stage because it has the smallest cold-side thermal capacity and the highest thermal conductance.

Fig. R2 a. COMSOL-simulated temperature evolution on thermoelectric surfaces of three devices (C-TED, CE-TED, CE-h-TED) under the driving current of 1.3 A; **b.** Infrared images of device surfaces under varying driving currents for C-TED, CE-TED, and CE-h-TED as reported in the original text.

Our COMSOL simulations in **Fig. R2a** also verify the explanation for this phenomenon. During the initial cooling phase (especially within the first 10 seconds), the temperature drop-rate of the heatsink-equipped CE-h-TED is significantly slower than that of the heatsink-free devices (C-TED). This delayed transient response explains why C-TED exhibits the lowest temperature at the 0.5 A measurement point captured at about 10s in Fig. 2g (here in Fig. R2b). Related explanations have been added to the main text and Supplementary Information as below.

- **Revision**

On page 9: "...It should be noted that all photographs were captured at 10s. As can

be seen from Fig. 2g, the C-TED cools faster than the heat sink-equipped CE-h-TED, resulting in the C-TED exhibiting a lower T at 10s. In contrast, the CE-TED demonstrates less effective cooling than the C-TED due to the additional introduction of EVA thermal resistance material on top of the C-TED structure, which can be well-explained by the heat transfer principles (see Figure S7-1 and S7-2 for further details)...”

9) Similarly in Fig. 2g at 1.3 Amp - C-TED appears cooler than CE-TED. Why so? Understand why CE-h-TED is the coolest.

Response:

We appreciate the reviewer for your insightful query regarding the temperature hierarchy in Fig. 2g at 1.3A. As measured in Fig. 2g, the surface temperatures captured at 10 s after the turning on of the cooling device follow $T_{CE-h-TED} < T_{C-TED} < T_{CE-TED}$, which is consistent with our predictive curves in Fig. R2a under the driving current of 1.3 A (Question 8). At the 10-second measurement point, the C-TED and CE-TED had completed their primary cooling phase, with CE-h-TED becoming the coolest owing to its lower T_h and G value. We will analyze this issue using Equation 1 from the previous question.

The lowest temperature that can be achieved by Peliter cooling ($T_{c,min}$) of a thermoelectric cooler (TEC) is determined by the steady-state heat balance. When the heat absorption at the cold junction (Peltier effect) balances with the heat dissipation capability at the hot side, heat leakage, and Joule heating, $dT_c/dt=0$, and the cold-side temperature stops decreasing. The steady-state equation is:

$$\alpha IT_c - \frac{1}{2} I^2 R = G(T_h - T_c) \text{ (Eq.8)}$$

Then, the expression for minimum temperature (neglecting environmental heat exchange) is:

$$\Delta T_{max} = T_h - T_{c,min} = \frac{\alpha IT_c - \frac{1}{2} I^2 R}{G} \text{ (Eq.9)}$$

From equation 9, we can conclude that $T_{c,min}$ is inversely related to G (**smaller G is better for deep cooling**), albeit at the cost of slower cooling speed, and reducing T_h is the key to breaking the limit of $T_{c,min}$. The device CE-h-TED has a smaller G value than C-TED (previously explained in question 8 with the result of $G_{C-TED} > G_{CE-TED} = G_{CE-h-TED}$) and smaller T_h when compared with CE-TED (Metal heat sink significantly lowers T_h), resulting in the smallest $T_{c,min}$ value within these three devices when reaching the steady state. Relevant explanations have been added to the main text and Supplementary Information as below.

- **Revision**

On page 9: “...In contrast, the CE-TED demonstrates less effective cooling than the C-TED due to the introduction of EVA thermal resistance material on top of the C-TED structure, which can be well-explained by the heat transfer principles (see Figure S7-1 and S7-2 for details). Under elevated driving currents (>0.9 A), all three devices exhibit accelerated cooling rates, achieving their respective minimum temperatures earlier. Consequently, the CE-h-TED demonstrates the most substantial temperature depression at high currents due to its superior heat dissipation capacity.”

10) Basically, the relative value of EVA vs Heat sink (HS) is not clear from data; please provide in the supplementary section.

Response:

We thank the reviewer for this constructive suggestion. The comparative parameters of EVA and Heat Sink (HS) are provided in **Table R1**.

Table R1. Relative value of EVA vs Heat sink.

	Size (L × W × H)	Thermal Conductivity (W/m·K)
EVA Film	80 mm × 60 mm × 2 mm	anisotropic mean: 0.28 ± 0.01
Copper Heat Sinks	9 mm × 9 mm × 3 mm (9 × 9 mm base with 16 fins of 2 mm height)	400

More material properties are added in the **Method Section** as below.

- **Revision in Main Text**

On page 8: “As shown in Figure 2b and S7, the EVA film features a highly porous structure, accounting for its low in-plane thermal conductivity of 0.267 W/m·K (longitudinal) and 0.271 W/m·K (transverse) at 313.15 K.”

- **Revision in Methods**

On pages 16-17: Materials. “...and EVA film from Dongguan Henglita Packaging Products Co. was laser-cut to match the CE-TED and CE-h-TED surface dimensions (thickness: 2 mm, area: 60 × 80 mm). The 48 copper heat sinks (9 × 9 mm base with 16 fins of 2 mm height) for CE-h-TED were provided by Chongqing Fengreng Feifei Electronic Technology Co., and 8 aluminum alloy heat sinks along with one driving PCB heat sink (20 mm × 15 mm × 5 mm, L × W × H) were brought from Shenzhen Maoyuan Heat Dissipation Technology.”

11) Line 153-154 - I like the concept of EVA as AR-coat and heat-spreading material in

plane; please provide thermal conductivity data for EVA in the plane of the film in the supplementary section.

Response:

We thank the reviewer for recognizing the dual functionality of EVA. The in-plane thermal conductivity data of EVA are now fully documented in **Supplementary Fig. S7a** (shown below). Results show that the in-plane transverse thermal conductivities are 0.267 W/(m·K) at 40°C, 0.284 W/(m·K) at 60°C, 0.273 W/(m·K) at 80°C, and in-plane longitudinal thermal conductivities are 0.271 W/(m·K) at 40°C, 0.284 W/(m·K) at 60°C, 0.305 W/(m·K) at 80°C.

Fig. S7a Vertical and lateral thermal conductivity of EVA sponge layer.

More material properties are added in the **Method Section** as below.

● **Revision in Main Text**

On page 8: As shown in Figure 2b and S7, the EVA film features a highly porous structure, accounting for its low thermal conductivity of 0.267 W/m·K (longitudinal) and 0.271 W/m·K (transverse) at 313.15 K.

12) Line 173-174 - the description of retention time is interesting - retention is better with heat-sink, understandably, and that means that set of data are different from (9) above; looks like we need Heat sink for temporal stability. This should be discussed in the summary/discussion. Also, did they use a fan in addition to the heat sink - this should be clearly noted; how was the heat dissipated from the heat sink?

Fig. R3 Surface temperature of C-TED, CE-TED, and CE-h-TED under identical driving conditions (0.1 A-1.7 A, 0.4 A in step, 25.4 °C ambient) measured via thermocouple.

Response:

We appreciate the reviewer's insightful observations regarding heat sink effects on retention time. In response to this question, here, we re-measured the surface temperature for all three types of devices under varying currents (Fig. R2) using a thermocouple. As shown in Fig. R3c, heat sinks significantly prolonged cooling retention, validating their necessity for temporal stability. We have included this finding in the **Conclusion** part of the manuscript.

Additionally, Figure 2g in the main text shows that $T_{\text{CE-h-TED}} < T_{\text{C-TED}} < T_{\text{CE-TED}}$ when the current was 1.3A. While transient cooldown is slower in CE-h-TED due to thermal mass, its ultimate surface temperature is lower than that of C-TED (confirmed by a darker thermal image under an IR camera), with heat sinks critically enabling superior steady-state performance under high loads.

All experiments were strictly conducted via natural convection without auxiliary fans, and we have explicitly noted in the revised Methods section, as copied below.

● **Revision**

On page 16: "...the active thermal stealth device achieves rapid response times (on the order of seconds), broad operational compatibility across both cold and hot environments (spanning 5.77 °C to 109.16 °C), and extended cooling duration enabled by integrated heat sinks..."

On page 17: "**Materials characterization and device performance measurement**...All experiments were strictly conducted via natural convection without auxiliary fans."

13) Fig. 3e - is the applied current 0.1 A? or is it much higher? Do we have similar displays for higher currents.

Fig. R4 Cooling display performance of the CE-h-TED under high current conditions.

Response:

Yes, Fig. 3e used a 0.1A drive current, and we have now added higher-current demonstrations in **Fig. R4** (also added in Supplementary Information Fig. S8). The results show that the full display runs smoothly up to 0.9A. Even at 0.9A, the thermal patterns stay uniform without distinct gaps between thermoelectric legs.

● **Revision**

On page 11: "...Figure 3e shows the cooling effect of the array device operating at a driving current of 0.1 A (demonstrations of higher current input were shown in Fig. S8),..."

14) Line 214-215 - at 1.9 A, $T_c = 5.77C$, $T_h = 109.8C$; $T_{amb} = 25C$. With the ambient being 25C and T_{cold} being significantly low at 5.77C - how was the condensation of atmospheric moisture avoided in the data? Please include these details in the Methods section.

Response:

Fig. R5 a. Optical image showing the experimental setup during device testing under normal operating conditions; **b.** Observation of condensation formation on the device surface when the ambient temperature is maintained around 5.8 °C for an extended period.

We thank the reviewer for this critical observation regarding condensation control. During the initial testing, we did not consider the issue of condensation. The cooling image display experiment in the manuscript was performed above 6 degrees, so we indeed did not observe any condensation phenomenon. Our tests confirm that devices exhibit surface condensation below 6 °C (with a relative humidity (RH) of ~60%) (**Fig. R5a** and R5b) by removing the EVA layer. At this preliminary research stage, condensation does not compromise electrical safety or IR stealth performance, but it should be carefully avoided in future applications. We have added this clarification to

the **Conclusion**.

● **Revision**

On page 16: "...Future implementations targeting low T operation should incorporate additional anti-condensation strategies...."

15) Line 225-227 - the authors mention passive cooling - it is not clear what they refer to.

Response:

Here, the *passive cooling* specifically refers to emissivity-modulated thermal regulation (e.g., VO₂ phase-change materials)^{1,2} and radiative cooling coatings^{3,4}, which rely on intrinsic material properties without external energy input. In contrast, thermoelectric-based temperature regulation achieves direct thermal changes through an external power supply, constituting *active cooling*. **We have revised the manuscript to explicitly state the difference.**

● **Revision**

On page 11: "In comparison, other thermal regulation strategies (e.g., active electrical T control, such as graphene fabric film²⁶ or passive thermal regulation emissivity-tuning material²⁷) typically exhibit response times on the order of minutes, making the second-level response of thermoelectric devices more suited to practical applications...."

Reference:

1. Xiao L., Ma H., Liu J., et al. Fast Adaptive Thermal Camouflage Based on Flexible VO₂/Graphene/CNT Thin Films. *Nano Lett.* 15, 8365-8370 (2015).
2. Li M., Cheng Y., Fang C., Xi, W., et al. W/Al Co-doping VO₂ nanoparticles for high performance passive infrared stealth films with enhanced durability. *Ceram. Int.* 50, 1443-1451 (2024).
3. G. Huang, A.R. Yengannagari, K. Matsumori, et al. Radiative cooling and indoor light management enabled by a transparent and self-cleaning polymer-based metamaterial. *Nat. Commun.* 15, 3798 (2024).
4. X. Zhang, Z. Cheng, D. Yang, et al. Scalable Bio-Skin-Inspired Radiative Cooling Metafabric for Breaking Trade-Off between Optical Properties and Application Requirements. *ACS Photonics* 10, 1624 (2023).

16) Line 360-361 - the *Methods* section is very terse; need more description - including

covering many of the above points

Response:

We thank the reviewer for emphasizing the need for methodological depth. Accordingly, we have significantly expanded the **Methods section** to incorporate: (1) EVA material and heat sinks specifications; (2) Kirigami fabrication details; (3) thermal conductivity of the EVA film; and (4) Measurement method as below.

● **Revision (On pages 16-17)**

Methods

Materials. N-type $\text{Bi}_2\text{Te}_{2.7}\text{Se}_{0.3}$ and P-type $\text{Bi}_{0.5}\text{Sb}_{1.5}\text{Te}_3$ ingots ($1.4 \text{ mm} \times 1.4 \text{ mm} \times 2.5 \text{ mm}$, $L \times W \times H$) were purchased from Wuhan SAGREON Co., and EVA film from Dongguan Henglita Packaging Products Co. was laser-cut to match the CE-TED and CE-h-TED surface dimensions (thickness: 2 mm, area: $60 \times 80 \text{ mm}$). The 48 copper heat sinks ($9 \times 9 \text{ mm}$ base with 16 fins of 2 mm height) for CE-h-TED were provided by Chongqing Fengreng Feifei Electronic Technology Co., and 8 aluminum alloy heat sinks along with one driving PCB heat sink ($20 \text{ mm} \times 15 \text{ mm} \times 5 \text{ mm}$, $L \times W \times H$) were brought from Shenzhen Maoyuan Heat Dissipation Technology....

Materials characterization and device performance measurement. ... The thermal conductivity of the EVA film was measured by the Laser thermal conductivity meter (LFA467) from NETZSCH. ... Surface temperature of C-TED, CE-TED, and CE-h-TED were also evaluated via thermocouple measurements (K-type, 1 mm diameter, $\pm 0.5^\circ\text{C}$ accuracy) in Supplementary information (Fig. S22). ... All experiments were strictly conducted via natural convection without auxiliary fans.

17) Line 351-352 - I was looking to see more data with their Kirigami approach and how it was implemented. Supplementary S1 and S2 do not appear sufficient. Basically, it would be good to know how "flexibility" is achieved

Response:

Figure. R6 a. Fabrication method of the kirigami structure. **b.** Stress distribution for

devices with/without Kirigami structure

We sincerely appreciate the reviewer's request for deeper methodological insights into our Kirigami approach. To address this, we have expanded Supplementary S2 with implementation details. The Kirigami patterning was achieved by precisely marking cut lines on the PI surface (positions mapped in **Fig. R6a**), followed by manual incision along these guides with a surgical scalpel. Flexibility enhancement originates from stress release via this kirigami structure, as validated by COMSOL simulations comparing devices with/without Kirigami (**Fig. R6b**). At 15° bending, peak stresses on the PI surface were 297 MPa (uncut) versus 60.4 MPa Kirigami-structured), demonstrating a 79.7% reduction and confirming the mechanism's efficacy.

Revisions to Fig. S2 in the Supplementary Information are shown below, and Fig. R6b has been added to Fig. S8 in the Supplementary Information.

Fig. S2 Fabrication flow chart diagram of the device.

● **Revision**

On page 2 of the Supplementary Information: “To achieve device flexibility, the Kirigami patterning was achieved by first precisely marking cut lines on the PI surface, followed by manual incision along these guides with a surgical scalpel.”

18) While IR-imaging data is the most relevant, given the targeted application and theme of the paper, some additional quantitative cooling data for the three cases (C-TED, CE-TED and CE-h-TED) would be good to show that - in all cases, the images are due to emissivity control indeed and not from any actual temperature differences; some limited thermocouple measurements on control test structures for each of the three

cases would be convincing.

Fig. R7 Surface temperature tested via thermocouple measurements of C-TED, CE-TED and CE-h-TED under identical driving conditions (0.1A-1.7 A, 0.4A in between, 25.4°C ambient)

Response:

We thank the reviewer for this essential suggestion to quantitatively decouple emissivity and temperature effects. We conducted additional thermocouple measurements (K-type, 1 mm diameter, $\pm 0.5^\circ\text{C}$ accuracy) on the surfaces of all three devices (**C-TED**, **CE-TED** and **CE-h-TED**) under identical driving conditions (0.1A-1.7 A, 0.4A in step, 25.4 °C ambient). As shown in **Fig. R7**, the tested cooling trends for all three devices aligned well with the data we tested via the IR camera. This confirms that the observed camouflage effects originate from thermal gradients instead of the emissivity control. Here, it needs to be explained that when the thermocouple is placed outside the EVA film layer for testing, the cooling effect measured outside the film is slightly worse than that measured by the infrared camera. We believe that the poor cooling effect obtained from the thermocouple testing is due to the following reasons: during the thermocouple measurement, it was placed on the outer surface of the porous EVA layer, whereas during the infrared camera testing, it was possible to detect part of the temperature from the cold end of the thermoelectric device below through the holes in the EVA layer, and the temperature at the cold end of the device's PI layer is even lower.

Raw data is added in Supplementary Information as Fig. S22, and related description has been added to the “Materials characterization and device performance measurement”.

● Revision

On page 17: “...Surface temperatures of C-TED, CE-TED, and CE-h-TED were also measured via thermocouple (K-type, 1 mm diameter, $\pm 0.5^\circ\text{C}$ accuracy), as shown in Supplementary information (Fig. S22).”

19) Finally the aspect of how much power and energy are used for the TED-based IR-image control, along with weight of TE modules and heat-sink, should be described in the main paper and with some specific data perhaps in the supplementary section - this will be important when readers look at the TED approach to competing approaches using meta-surfaces and phase change materials.

Table R2. Power consumption for the TED-based system.

	1	2	3	4	5	6	7	8	9	10
Applied Voltage (V)	3	3	3	3	3	3	3	3	3	3
Applied Current for Whole System (A)	0.38	0.59	0.78	0.98	1.18	1.36	1.57	1.77	1.95	2.14
Power Consumption (W)	1.14	1.77	2.34	2.94	3.54	4.08	4.71	5.31	5.85	6.42

Table R3. Weight for C-TED, CE-TED, and CE-h-TED.

	C-TED	CE-TED	CE-h-TED
Weight (g)	32.24	32.48	93.20

Response:

We thank the reviewer for emphasizing the practical metrics of our TED-based system. Comprehensive power consumption for the whole TED-based system and weight data for three types of devices are summarized in **Tables R2 and R3**, respectively. Specific data were also added to the supplementary information in **Tables S1 and S2**. Related discussions on power and weight were discussed in the main text.

● **Revision**

On page 5: "...Notably, this performance is sustained across an operating power range of 1.14 W to 6.42 W, corresponding to driving currents of 0.38 A to 2.14 A, respectively (detailed power consumption profiles and their estimated battery life are listed in Tables S1 and S2). Circuit implementation strategies enabling this performance are further illustrated in Supplementary Figures S4 to S6.'

On pages 9-10: "...Device mass measurements (Table S3) reveal distinct weight profiles: C-TED (32.24 g), CE-TED (32.48 g), and CE-h-TED (93.2 g). The EVA film contributes a marginal 0.24 g mass increment, whereas the heat sink introduces a substantial 60.72 g addition. This mass penalty is counterbalanced by the heat sink's critical role in enabling significant thermal display enhancement."

Reviewer #2:

This manuscript introduces a multifunctional infrared (IR) stealth cloak built from a thermoelectric device (TED) array capable of active camouflage, deception, and messaging. Leveraging fast, programmable heating and cooling via the Peltier effect and integrated circuit control, the authors achieve dynamic temperature shaping with remarkable uniformity, rapid response (~2 seconds), and wide operation temperatures (5.77 °C to 109.16 °C). Key innovations include the use of a porous EVA film to enhance temperature uniformity and a kirigami-structured flexible PCB enabling wearable applications. The work represents a substantial advancement in the field of active thermal stealth and adaptive IR display technologies.

The main strengths of the presented work include an operational temperature range that is broader than previously reported systems of its kind and a response time (~2 seconds) that is significantly faster than conventional materials like VO₂-based systems, which operate over minutes. Moreover, the programmable resolution with low-temperature standard deviation (~0.19 K) per pixel demonstrates very good thermal uniformity. The use of porous black EVA film reduces surface reflectivity from ~32% to ~3%, significantly improving stealth performance.

The authors also utilize clever circuit design (PWM, low-resistance driver chips, copper coverage) to minimize parasitic heating and support spatially resolved thermal control and incorporate kirigami patterns to improve mechanical adaptability without sacrificing function, enabling wearable applications.

In general, the quality of the manuscript and the level of the shown results are consistent with the publication in Nature Communications.

Response:

We sincerely appreciate your recognition of our work. In direct response to your insightful suggestions, we have conducted new experiments validating the device's real-world reliability, with all supporting data discussed in the main text or Supplementary Information (Figs. S15-S18). Your constructive feedback continues to guide our refinements, and we remain committed to addressing every point raised to enhance the manuscript's quality.

1) High functionality is achieved, but it is unclear if the power demands, up to 1.9 A per panel at 3V, could limit battery-powered, portable, or long-duration applications. The authors should provide a quantitative discussion of energy usage over time and thermal-to-electrical efficiency.

Table R4. The estimated power consumption and its battery life under different operating modes.

Operation Mode	System Current (A)	Power (W)	Energy/Cycle (J)	Endurance
Active Mode (~1.9A for single pixel)	2.14	6.42	~38.52	~1.15 hours
Baseline Stealth (~0.1A for single pixel)	0.38	1.14	~3.42	~6.49 hours

Response:

We thank the reviewer for raising this critical practical consideration. The system operates within a **power range of 1.08–5.64 W** (0.36–1.88 A @ 3V DC), with complete energy profiles detailed in **Supplementary Table S1 (summarized in Table R1)**. While peak power states (e.g., 6.42 W at 2.14A/3V) present challenges for continuous battery operation, the following strategies ensure viable portable and long-duration deployment:

1) Active high-power mode: Active camouflage requires only **6 s per operation** at 1.9 A (Fig. 3a), consuming **38.52 J/cycle**. This enables **>690 cycles (~1.15 h)** on a single 18650 cell (3.7V, 2000mAh, 7.4 Wh)- compact enough ($\Phi 18 \times 65$ mm) for portable integration.

2) Low-power Mode: Maintaining baseline stealth at **1.14 W** (0.38 A) extends operational endurance by ~5 times compared to sustained high-power modes.

Fig. R8 COP of the CE-h-TED Device under Electric Driving in Cooling/Heating Display Mode Based on Simulation

For thermal-to-electrical efficiency, we believe “thermal-to-electrical” conversion is not applicable here as TEDs consume electricity to manipulate thermal signatures. We believe you might be asking about **electrical-to-thermal efficiency** instead. To address this, we simulated the Coefficient of Performance (COP) for the CE-h-TED in both

cooling and heating display modes (**Fig. R8**). The maximum COP values reached **2.76 (cooling)** and **-3.07 (heating)**—the negative sign indicates reversed heat flow direction in heating mode. It should be noted that these are ideal simulation results; actual COP values may be slightly lower due to contact resistance in real-world operation.

● **Revision**

On page 5: "...Notably, this performance is sustained across an operating power range of 1.14 W to 6.42 W, corresponding to driving currents of 0.38 A to 2.14 A, respectively (detailed power consumption profiles and their estimated battery life are listed in Tables S1 and S2)..."

2) The device uses a 6×8 array (48 pixels), with each pixel consisting of 8 thermoelectric pairs. While effective for proof-of-concept, larger arrays or finer resolution for higher fidelity deception or messaging would likely require significant increases in complexity and power. Some discussion on scaling strategies would be helpful.

Response:

We appreciate the reviewer for raising this significant and insightful point, which correctly identifies the scalability limitations of our current 6×8 array (48-pixel) device and the inherent complexity/power challenges in pursuing larger arrays or finer resolutions for high-fidelity deception/messaging. We fully agree that discussing viable scaling strategies is crucial for evaluating the technology's future potential.

Indeed, larger arrays or finer resolution would: (1) intensify thermal crosstalk between adjacent pixels due to reduced pitch^{1,2}, and (2) exacerbate control signal routing, drive circuit complexity, and thermal interference^{3,4}. To address these, future work will explore compact pixel structures with enhanced thermal isolation, materials with higher thermoelectric efficiency (ZT values), and modular designs partitioning the array into smaller independently addressable/driven submodules (e.g., multiple 6×8 or larger units). We have incorporated an expanded discussion on these scaling challenges and strategies in the revised Conclusion section.

Reference:

1. Ouyang H, Gu Y, Gao Z, et al. Kirigami-inspired thermal regulator. *Physical Review Applied*, **19**, L011001 (2023).
2. Zhao Y, Liang Q, Li S, et al. Thermal Emission Manipulation Enabled by Nano-Kirigami Structures. *Small*, **20**, 2305171 (2024).
3. Park Y J, Song G, Shin J, et al. Shape memory alloy-based tensile activated kirigami

actuators. *Composites Part B: Engineering*, **305**, 112757 (2025).

4. Rafsanjani A, Jin L, Deng B, et al. Propagation of pop ups in kirigami shells. *Proceedings of the National Academy of Sciences*, **116**, 8200 (2019).

● **Revision**

On page 16: "...While the kirigami design provides some flexibility, the device's deformability remains constrained. Scaling to larger arrays or higher resolutions faces significant challenges, including intensified thermal crosstalk and heightened drive complexity³³⁻³⁶; Future work will therefore focus on two parallel tracks: (1) addressing scalability through architectures design, advanced thermoelectric materials (high ZT), and pixel-level thermal isolation designs to enable practical deployment, and (2) innovating thermoelectric materials and structures (e.g., thin-film TED²⁰) to develop thinner, more flexible devices—ultimately paving the way for infrared stealth cloaks that truly resemble invisibility cloaks. We believe this work holds revolutionary potential for IR thermal stealth technologies in surveillance and broader societal applications."

3) *While the kirigami structure allows bending and flexibility, long-term mechanical fatigue, delamination, and thermal cycling stability were not explored in detail. Did the authors test fatigue (e.g., >1,000 bend cycles) and resistance to environmental conditions (e.g., humidity, dust)?*

Response:

Fig. R9 Flexibility and stability testing of the IR-TED. a. Photograph of C-TED

attached to a curved surface. **b.** The internal resistance of eight regions randomly chosen from each row of the TED panel. **c.** The resistance variation of the eight-pixel region when the TED panel was bent in different bending radii (R is from 2 cm to 4 cm with 0.5 cm in between). **d.** The resistance variation of eight-pixel regions for 10000 bending cycles with a bending radius of 2 cm.

We sincerely appreciate your insightful comments regarding the long-term mechanical and environmental robustness, which we believe is vital for wearable applications.

For CE-TED with good flexibility, **fatigue testing under 10,000 bending cycles** was conducted (now explicitly shown in Fig. R9d, originally in Supplementary Fig. S9), demonstrating exceptional stability with $\leq 5.79\%$ maximum internal resistance changes across all eight regions (Fig. 4c and d).

Fig. R10 Imaging Performance of CE-h-TED in varied dust concentrations. (a) Air condition tester in ambient air; (b–f) Functional demonstration under increasing dust levels (PM2.5, PM1.0 and PM10) within a half-sealed chamber.

For the device’s resistance to environmental conditions, supplementary humidity and dust resistance tests were conducted (Fig. R7-R8). For dust testing, the device was placed in a half-sealed acrylic chamber (38 cm \times 38 cm \times 38 cm, L \times W \times H) with an IR camera port. Standardized electronic-grade artificial dust (sourced from Zhongwei Instruments) was blown inside the chamber incrementally. Results (**Fig. R10**) confirm normal thermal operation at all dust concentrations, though image clarity gradually decreased at higher dust concentrations. This might be due to IR scattering. Future optimization efforts will prioritize the adoption of advanced packaging to resist

the interference of particulate matter, thereby ensuring the clarity of images in harsh environments.

Fig. R11 Imaging performance of CE-h-TED under humidity gradients. **a.** Humidity test chamber setup; **b-f.** Functional demonstration across varying relative humidity (RH) levels with sustained thermal display fidelity.

For *humidity testing* (Fig. R11a), a humidifier increases chamber humidity levels from 50% to 90% RH. The device maintained full functionality across this range (Fig. R11b-f). Notably, no electrical shorts occurred even at 90% RH, where visible condensation formed on chamber walls, with thermal images remaining clear.

The above-mentioned content has been fully incorporated into the Supplementary Information (Figs. S16 and S17) and summarized in the main text.

● **Revision**

On page 14: “...Comprehensive supplementary validations confirm exceptional operational resilience, including reliable self-concealment functionality (Fig. S14), stable information integrity across thermal environments (Fig. S15), effective humidity resilience (Fig. S16), uncompromised performance under low-dust conditions (Fig. S17), and robust light-interference resistance (Fig. S18), collectively demonstrating the device's field-deployable robustness within the Supplementary Information.”

4) *How would the proposed system perform under varying ambient radiation (e.g., sunlight, rain) or dynamic backgrounds (changing temperature, clouds)? This would be relevant for the eventually sought applications.*

Response:

Thanks for bringing up this super practical question, it's exactly what we need to tackle

for real-world use!

Fig. R12 Heating/Cooling performance of the device surface under different light radiation.

First, for *sunlight changes*, we have bought an adjustable sun simulator lamp that covers all light spectra and turned it on from total dark up to bright, with the brightness ranging from 0 to 14.5 W/m^2 . As shown in **Fig. R12**, the heart-shaped image, both in cooling and heating performance, could be clearly demonstrated on the panel. Super bright light (high radiation) does heat the whole testing area (more experiment details are shown in *ambient temperature effects* below).

For the *rainy* condition, since hosing down our electronics with water risked shorts (those driver boards and power source aren't waterproof yet), we didn't simulate heavy downpours. But from our max-humidity tests (90% RH, shown in **Fig. R11** of Question 3, light rain should be totally fine. In the future, we'll switch to battery packs and seal everything up properly for rainy conditions.

Fig. R13. Thermal information display under varying ambient temperatures. a.

Experimental setup; **b-f**. Functional demonstration of the “YES” character display across progressively increasing thermal environments.

For *dynamic background* testing, we introduced hot air into the chamber to elevate the ambient T (T_{amb}) from 28°C to 50°C (**Fig. R13a**). Results in Fig. R13b-f demonstrate that as T_{amb} increased, the cooled “E” in our “YES” demo actually became more visible from Fig. R13b to c (hotter background means better cool-spot contrast!). As the T_{amb} kept going up, the whole image shown on the display became hotter, with the heated “YS” turning brighter and the cooled letter “E” becoming brighter.

Although T_{amb} influences device performance, we believe this can be mitigated through adaptive feedback controls that dynamically adjust display parameters to amplify spot contrast for messaging mode. The above-mentioned content has been fully incorporated into the Supplementary Information (Figs. S15 and S18) and summarized in the main text.

- **Revision**

On page 14: “...Comprehensive supplementary validations confirm exceptional operational resilience, including reliable self-concealment functionality (Fig. S14), stable information integrity across thermal environments (Fig. S15), effective humidity resilience (Fig. S16), uncompromised performance under low-dust conditions (Fig. S17), and robust light-interference resistance (Fig. S18), collectively demonstrating the device's field-deployable robustness within the Supplementary Information.”

5) Same as above for wearable applications: a person engaged in intense physical activity could modify the heating of the body. Could that affect the performance of the display?

Fig. R14. On-Body thermal display under different body temperatures. Image of

“RT” letter (Running Test for short) shown on the bending display (CE-TED) under **a.** Pre-exercise state; **b.** Post-3min fast treadmill running state; and **c.** Postprandial metabolic elevation state.

Response:

Here, we employ the bendable CE-TED for the following tests. In **Fig. R14**, the volunteer was wearing the CE-TED on his wrist, and the device performance of the letter “RT” (“Running Test” for short) was tested under the pre-exercise state and post-exercise state. After 3 min of fast treadmill running, the wrist skin temperature increased slightly (+0.9°C vs. pre-exercise state in Fig. R14a) with no distinct degradation in “RT” character recognition. Additionally, after dinner, a significant wrist temperature increase of 4.05°C (reaching 36.1°C) was observed in Fig. 14c, yet the device maintained full functionality, achieving complete recognition of the test letters. These findings indicate that within natural thermophysiological ranges (fluctuations <5°C), CE-TED performance is not significantly affected by epidermal temperature variations, ensuring reliable operation during human activities.

The above content has been fully incorporated into the Supplementary Information (Figs. S21) and mentioned in the main text.

● **Revision**

On page 15: “...Further wearable examples combining both heating and cooling in a bent state are provided in Fig. S20 and Fig. S21 of the Supplementary Information...”

6) Has the approach been tested with adversarial detection models to quantify how well the system can evade modern IR tracking or recognition software? Is there any operation metric (beyond what is reported in Fig. 1f) that allows us to establish how well this system can perform in practice? This would give objective insight into effectiveness beyond controlled lab conditions.

Response:

We sincerely appreciate your insightful suggestion to evaluate system effectiveness against adversarial IR detection models, which is critical for assessing real-world stealth capability that we regrettably did not address in the current study. At this stage, our validation on thermal stealth fully relied on thermal imaging via the FLUKE Ti489 PRO (thermal sensitivity of 0.03°C @30°C with 172800 thermal sensation pixels). We fully acknowledge that testing with military-grade recognition algorithms would provide indispensable metrics like evasion probability and counter-scannability. Given the specialized resources required (access to classified detection software/field testing),

this remains challenging for us now, but we commit to pursuing such validation in future collaborations with defense partners, and will incorporate adversarial testing protocols alongside practical operation metrics (e.g., false-positive rates under environmental noise) in our next-phase research to bridge the lab-to-field gap.

7) *Other minor comments:*

- *I could not see where EVA is defined.*

- *The analogy to Harry Potter's invisibility cloak is engaging but should be used with care in formal communication to avoid undermining the technical rigor.*

Response:

We sincerely thank the reviewer for these insightful observations. Regarding the EVA definition, we have now explicitly defined Ethylene-Vinyl Acetate (EVA) at its first occurrence in the Abstract and the main text to ensure clarity. For the Harry Potter analogy, we fully agree with the need to maintain technical rigor and have replaced the cultural reference with the revised statement: "The proposed IR stealth device is designed specifically to conceal objects from infrared cameras, not visible light", thereby eliminating informal comparisons while preserving the core technical distinction. These updates align with formal scientific discourse as suggested.

Reviewer #3:

We sincerely thank both the primary reviewer and the co-reviewing early-career researcher for their time and expertise through Nature Communications' peer review training initiative. We have responded to every question raised in your joint report and remain fully committed to incorporating any additional feedback to meet the journal's high standards.

Authors' response to the reviewers

Reviewer #1 (Remarks to the Author):

Thanks again to the authors for providing a continuously numbered revised text for an easier review as well as for addressing my suggested edits/clarifications.

Here are some suggested edits:

1) Line 58-59: instead of saying "effective in specific bands" change to "effective in narrow bands" as that's the focus of that section. Specificity is not the problem; it's the narrowness of the band, that's being addressed.

Response: Thank you for this insightful suggestion. We agree that emphasizing the "narrowness" of the bands more accurately captures the limitation we intended to describe. We have revised the sentence accordingly. The phrase "**effective in specific bands**" has been changed to "**effective in narrow bands**" in the revised manuscript.

2) Line 59-60: could be changed to "due to their resonant design constraints limiting the enablement of general applicability across a broad IR spectrum".

Response: We thank the reviewer for this valuable comment, which helps clarify the nature of the technical limitation. We have modified the sentence as suggested to better explain the constraint. The clause "**due to their resonant design constraints**" has been expanded to "**due to their resonant design constraints limiting the enablement of general applicability across a broad IR spectrum**".

● **Revision**

On page 2: "...While effective in narrow bands (e.g., MWIR/LWIR), these methods face inherent bandwidth limitations due to their resonant design constraints limiting the enablement of general applicability across a broad IR spectrum."

3) Line 64-66: Low-temperature scenarios can mean many things and the authors need to be more specific. Could they say, "Particularly in near-ambient scenarios, ϵ -focused strategies alone cannot achieve stable stealth without active cooling and/or heating that is enable by thermoelectric devices".

Response: Thank you for your insightful comment regarding the need for more specificity about temperature scenarios. We agree that the original term "low-temperature" was too broad. We have revised the sentence accordingly in the manuscript.

In our modification, we specifically adopted the phrase "**sub-ambient and near-ambient temperature scenarios**". We believe that including "**sub-ambient**" (below ambient temperature) is particularly important, as this scenario most distinctly highlights the critical advantage of thermoelectric devices—their ability to provide active cooling—which is a unique capability not addressable by passive ϵ -focused strategies alone. This change more accurately reflects the specific operational challenge and the unique solution our work discusses.

● **Revision**

On pages 2-3: "...Particularly in sub-ambient and near-ambient temperature scenarios, ϵ -focused strategies alone cannot achieve stable stealth without active cooling and heating that is enabled by thermoelectric devices..."

4) *Line 115-116 - thanks for the clarification and sticking with scientific terms, and avoiding reference to Harry Potter.*

Response: Thank you for your positive feedback. We truly appreciate your recognition of our effort to maintain scientific clarity and appropriate terminology in the manuscript.

5) *It is good that the authors have revised the supplementary text, with figures and tables being numbered as they appear first in the main manuscript.*

Response: Thank you for your thoughtful suggestion. We are glad to hear that the revised numbering of figures and tables in the supplementary material contributes to a clearer and more logical flow of the article. We truly appreciate your guidance in enhancing the overall coherence of the manuscript.

6) *Table R1 and Fig. S7a are understandable and sufficient for this paper, but I suggest that would be good for the authors to note that this EVA usefulness is to be studied further and other materials could be considered for such purposes. Note - this is only a suggestion.*

Response: Thank you for your valuable suggestion. We have added a note in the manuscript to highlight that similar thermal homogenization effects may also be achievable using other porous materials with high thermal resistance, and that further study in this direction could help broaden material choices and improve thermal management strategies.

● **Revision**

On page 9: "...It is worth noting that similar thermal management effects could also be achieved using other porous materials with high thermal resistance, which are expected to facilitate transverse heat spreading and suppress longitudinal conduction. Further exploration of such materials may broaden the applicability and optimization of thermal homogenization strategies.

7) *Thanks for the clarification of the non-use of auxiliary fans.*

Response: Thank you for your feedback. We are glad that the clarification regarding the non-use of auxiliary fans was clear and helpful.

8) *Line 259-264: the authors note the speed of cooling with their bulk TECs in seconds, relative to other thermal regulation strategies that take minutes. It is worth noting here, for the general readers, that thin-film thermoelectric devices can be much faster and also energy efficient than bulk TECs (see for example -Fig. 2g, Fig. 2h, Fig. 2i in reference <https://doi.org/10.1038/s41551-023-01070-w>); this also applies to Line 384-385. It would be worth for the readership to know what has been shown/published already.*

Response: Thank you for bringing this important point to our attention. We have carefully studied the reference you suggested (DOI: 10.1038/s41551-023-01070-w) and agree that it provides valuable context regarding the performance of thin-film thermoelectric devices. In particular, we noted that the response time reported in Figures 2g and 2h of that work is approximately 3 seconds, which is indeed comparable to the 2.71 seconds of response time achieved in our bulk device.

We have revised our manuscript to include a discussion of these findings, **noting that while the response time of our device is similar to that of advanced thin-film systems, the thin-film platform may offer advantages in energy efficiency.** The corresponding reference has been added in the section addressing device dynamic performance.

● **Revision**

On page 11: “...It is worth noting that such second-level response performance is comparable to state-of-the-art thin-film thermoelectric devices, which also typically exhibit cooling responses on the order of several seconds, while thin-film type potentially offering improved energy efficiency³³...”

Reference

33. Osborn, L.E., Venkatasubramanian, R., Himmtann, M. *et al.* Evoking natural thermal perceptions using a thin-film thermoelectric device with high cooling power density and speed. *Nat. Biomed. Eng* **8**, 1004–1017 (2024).

9) *Line 397-398: pixel-type thermal isolation and IR imaging has been shown with thin-film thermoelectric devices (see <https://www.nature.com/articles/s41467-025-59698-y>) over a large area. It would be worth for the readership to know what has been shown/published already.*

Response: Thank you for suggesting this highly relevant reference. We have now included a citation to the work in the revised manuscript at the appropriate location, as below. This addition helps acknowledge the important prior demonstration of large-area, pixel-type thermal isolation and IR imaging using thin-film thermoelectric devices, providing readers with a more comprehensive context for existing advances in the field.

● **Revision**

On page 16: “Future work will therefore focus on two parallel tracks: (1) addressing scalability through architectures design, advanced thermoelectric materials (high ZT), and pixel-level thermal isolation designs to enable practical deployment³⁸, and (2) innovating thermoelectric materials and structures (e.g., thin-film TED²⁰) to develop thinner, more flexible devices—ultimately paving the way for infrared stealth cloaks that truly resemble invisibility cloaks.”

Reference

38. Ballard, J., Hubbard, M., Jung, S.J. *et al.* Nano-engineered thin-film thermoelectric materials enable practical solid-state refrigeration. *Nat. Commun.* **16**, 4421 (2025).

The authors have done very interesting work and explored an important area in the IR emission control of surfaces and have described a variety of possibilities. Hence, I recommend the publication with the above suggested edits and clarifications.

Reviewer #2 (Remarks to the Author):

The authors satisfactorily addressed our comments. I particularly appreciated the experiments at different light intensities. It looks like typical daylight irradiation could hinder the proposed approach but, still, while further work will be needed to extended the operability of the proposed strategies, the shown advancement are relevant. I can recommend the publication.

Response: Thank you very much for your positive feedback, and your insightful comments throughout the review process have greatly strengthened the manuscript. We are especially grateful that you highlighted the experiments under different testing conditions as valuable additions to the manuscript. We agree that extending the operability of the proposed strategy under stronger daylight conditions represents an important direction for future work, and we truly appreciate your recognition of the relevance and significance of the advancements presented here.

Authors' response to the reviewers

Reviewer #1 (Remarks to the Author):

Dear Authors,

The edits to the paper look fine.

Regards!

Response: Thank you for your time dedicated to improving the quality of this manuscript. We are happy that this manuscript will be published with your support.